

# The propagator of the finite XXZ spin-$\frac{1}{2}$ chain

Gyorgy Z. Fehér[1*] and Balázs Pozsgay[1,2]

**1** Department of Theoretical Physics, Budapest University of Technology and Economics,
1111 Budapest, Budafoki út 8, Hungary
**2** BME Statistical Field Theory Research Group
1111 Budapest, Budafoki út 8, Hungary

⋆ g.feher@eik.bme.hu

## Abstract

We derive contour integral formulas for the real space propagator of the spin-$\frac{1}{2}$ XXZ chain. The exact results are valid in any finite volume with periodic boundary conditions, and for any value of the anisotropy parameter. The integrals are on fixed contours, that are independent of the Bethe Ansatz solution of the model and the string hypothesis. The propagator is obtained by two different methods. First we compute it through the spectral sum of a deformed model, and as a by-product we also compute the propagator of the XXZ chain perturbed by a Dzyaloshinskii-Moriya interaction term. As a second way we also compute the propagator through a lattice path integral, which is evaluated exactly utilizing the so-called $F$-basis in the mirror (or quantum) channel. The final expressions are similar to the Yudson representation of the infinite volume propagator, with the volume entering as a parameter. As an application of the propagator we compute the Loschmidt amplitude for the quantum quench from a domain wall state.

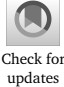

# 1 Introduction

Quantum integrable models are special theories that can describe strongly correlated many body systems, such that their Hamiltonians can be diagonalized using exact methods. Their study goes back to the solution of the Heisenberg spin chain by Bethe [1]. Whereas the largest part of the literature is devoted to the study of the state functions and correlation functions in the ground state or at finite temperatures [2–12], recently considerable interest was also devoted to the study of out-of-equilibrium situations (see [13, 14]). This is motivated by experimental advances that make it possible to measure the dynamical properties of isolated quantum systems [15–19], thus prompting for a theoretical understanding of the observed phenomena. Two particular areas that have been actively investigated in recent years are the equilibration/thermalization of isolated integrable systems, and the description of their transport properties.

Regarding equilibration the main paradigm is that of the Generalized Gibbs Ensemble (GGE) [20]: it is now widely believed that in homogeneous situations isolated time evolution leads to a steady state that is characterized by a complete set of (local and quasi-local) conserved charges of the model. The imprecise notion of the "complete set" of the charges can be defined rigorously by focusing on particular models; for example in the spin-1/2 XXZ chain the complete GGE was established using Bethe Ansatz techniques in [21]. Furthermore, in this model there are exact methods to compute the mean values of local observables in the highly excited states that emerge after the quantum quenches [22, 23]. On the other hand, the GGE has not yet been established for models with higher rank symmetries.

The second area, namely the description of transport in integrable systems has been treated

by the so-called Generalized Hydrodynamics (GHD) [24–28]. In the first approximation this theory captures the physics at the Euler-scale (a combined large time and long distance limit), and it rests on the dissipationless scattering in integrable models, guaranteeing ballistic propagation of the quasi-particles. However, it has been shown recently that diffusion can also be described within the same framework [27]. The GHD has been established for a number of models and it is extremely successful: it provides predictions that agree with DMRG calculations up to many digits [25, 27, 29]. The theory can also describe two-point or higher point correlation functions in the Euler scale limit [26].

It would be interesting to go beyond the GHD and derive exact results for time evolution in interacting integrable models. One the one hand, this could provide a microscopic derivation of the GHD. On the other hand, it could give access to physical effects beyond its reach. In the following we review some of the approaches towards exact treatment of non-equilibrium dynamics, with a special focus on the XXZ chain.

First of all, the direct method consists of the insertion of (one or two) complete sets of eigenstates in finite volume and of the computation of the resulting spectral series. Analogous computations have been performed in the equilibrium case regarding dynamical correlation functions in the earlier works [6,7], but there are fewer results available in the non-equilibrium case. Depending on the situation the problem can be treated numerically [30], or one can obtain analytic expressions for power-law correction terms in the long-time limit [31,32] using the Quench Action logic [33]. A necessary ingredient in any such calculation is to have exact formulas for the overlaps with the initial states; these are known in a number of cases [34–39]. It was also argued recently in [40] that one should focus on a sub-class of initial states (called integrable states) where factorized overlaps can be expected. We should note that for such states the time evolution of the von Neumann and Rényi entropies have been computed in [41–44].

An independent approach is that of the Quantum Transfer Matrix (QTM) method, which was originally devised to compute the thermodynamics of the spin chain [45]. The main idea is to build a lattice path integral for the partition function, which can be evaluated in the so-called quantum (or rotated, or mirror) channel by exchanging the space and time coordinates. This way the summation over all the eigenstates of the system is replaced by the focus on a single leading eigenstate of the QTM. The method was generalized in [46–48] to yield the Loschmidt amplitude in certain homogeneous quenches. It was already argued in [46] that even non-equilibrium time dependent correlators could be computed with the QTM, somewhat analogous to the determination of finite temperature static and dynamical correlators [49,50]. However, the computations have not yet been carried out and are expected to be considerably more involved.

A further idea towards exact treatment of real time dynamics is through the Yudson representation for the propagator of integrable models. Originally developed in [51,52] and worked out for the Lieb-Liniger and XXZ models in [53–56], this method computes the propagator of a finite number of particles in an infinite volume system. It is built on two basic ideas. First, it uses the fact that the Bethe wave functions form a complete set in infinite volume, and instead of a summation over the Bethe roots (solutions to the Bethe equations in finite volume) one needs to integrate over the rapidities with appropriate weight functions [57, 58]. Depending on the model one can have remarkable simplifications, for example in the XXZ chain or the Gaudin-Yang model the Yudson representation involves single integrals over certain contours instead of a sum of integrals over all string states [56,59]. The second idea is more technical: in the resulting multiple integral formula one can replace one side of the propagator (corresponding either to the "in" or the "out" configuration) by a free wave function, leading to a further considerable simplification. We should also note that the basic ideas of the Yudson representation appeared independently in [60].

It was already demonstrated in [54–56] that the Yudson method can yield concrete predictions for real time evolution of observables, nevertheless it has severe limitations. First of all, the number of integrals in the propagator is always equal to the number of particles involved. Thus it is quite difficult to take the physical thermodynamic limit. Second, it is an infinite volume method, therefore it can only describe physical processes where the particles disperse into infinity after some initial interaction.

In the present work we compute the propagator of the XXZ chain in finite volume. The advantage of our approach is that the finite volume propagator can describe both spatially homogeneous and inhomogeneous situations, and it could also be used to study finite volume effects. We use two different methods, and our derivations are independent from the works on the Yudson approach. First, we employ a direct spectral representation, building on the results on [6, 7]. Second, we also use the QTM approach to develop a lattice path integral for the propagator, which we evaluate exactly for any finite volume. Our final formulas are similar, but not identical to the Yudson representation.

The paper is organized as follows. In 2 we introduce the model and the real space propagator. In 3 we compute the propagator using the spectral sum, leading to a multiple integral formulas. In 4 we also develop a different method to compute the propagator, and present the results up to the two-particle case. In 5 we consider an application for the propagator: the Loschmidt amplitude for the quench from the so-called domain wall state. We conclude in 6, and some of the more technical calculations are detailed in the appendices B-C.3.

## 2 The propagator in the spin basis

We consider the spin$-\frac{1}{2}$ XXZ model, described by the following Hamiltonian:

$$H = \sum_{j=1}^{L} \left( \sigma_j^x \sigma_{j+1}^x + \sigma_j^y \sigma_{j+1}^y + \Delta \left( \sigma_j^z \sigma_{j+1}^z - 1 \right) \right). \tag{2.1}$$

Here $\sigma_j^x$, $\sigma_j^y$, $\sigma_j^z$ are the usual Pauli matrices, acting on the $j$th subspace of the tensor product space $\otimes_{j=1}^{L} \mathbb{C}^2$. We assume periodic boundary conditions

$$\sigma_{L+1}^x = \sigma_1^x, \qquad \sigma_{L+1}^y = \sigma_1^y, \qquad \sigma_{L+1}^z = \sigma_1^z, \tag{2.2}$$

and use the following parametrization of the anisotropy parameter $\Delta$:

$$\Delta = \cosh \eta. \tag{2.3}$$

We do not restrict ourselves to any regime in $\Delta$.

Our goal is to compute the real space propagator of the spin chain, which is defined as follows. First we choose the reference state to be the state with all spins up:

$$|0\rangle = \otimes_{i=1}^{L} \begin{pmatrix} 1 \\ 0 \end{pmatrix}_{[i]} = |\uparrow\uparrow\ldots\uparrow\rangle. \tag{2.4}$$

We denote the basis state with $m$ spins down at positions $a_j \in \{1,\ldots,L\}$, $j = 1\ldots m$ by

$$|a_1,\ldots,a_m\rangle = \prod_{j=1}^{m} \sigma_{a_j}^- |0\rangle. \tag{2.5}$$

For the coordinate variables we always assume $a_j < a_k$ for $j < k$.

The real space propagator is then defined as

$$G_m(\{b\}, \{a\}, t) = \langle b_1, \ldots, b_m | e^{-iHt} | a_1, \ldots, a_m \rangle. \tag{2.6}$$

Note that the in and out states have the same magnetization; the spin-$z$ conservation of the Hamiltonian implies that all other matrix elements of the propagator are identically zero. The propagator depends on the volume $L$, but for simplicity we omit this in the notation.

The propagator satisfies the Schrödinger-type equations

$$i\frac{d}{dt} G_m(\{b\}, \{a\}, t) = \hat{H}_a G_m(\{b\}, \{a\}, t) = \hat{H}_b G_m(\{b\}, \{a\}, t) \tag{2.7}$$

and the initial condition

$$G_m(\{b\}, \{a\}, 0) = \prod_{j=1}^{m} \delta_{a_j, b_j}. \tag{2.8}$$

In (2.7) $\hat{H}_{a,b}$ are operators that act as the Hamiltonian on the corresponding coordinates. The equality between the second and third expressions in (2.7) follow from the fact that the Hamiltonian is a symmetric matrix in the spin basis: it can be seen from (2.1) that all its matrix elements are real, and a real Hermitian matrix is symmetric.

In the following we discuss the symmetry properties of the propagator. It follows from the definition that

$$G_m(\{b\}, \{a\}, t) = G_m^*(\{a\}, \{b\}, -t). \tag{2.9}$$

However, a stronger condition also holds, the propagator is a symmetric matrix in the spin basis:

$$G_m(\{b\}, \{a\}, t) = G_m(\{a\}, \{b\}, t). \tag{2.10}$$

This follows from the second equality in (2.7), or alternatively, from the fact that the exponentials of the Hamiltonian are also symmetric.

Translational invariance and space reflection invariance lead to the conditions

$$G_m(\{b\}, \{a\}, t) = G_m(1 + \{b\}, 1 + \{a\}, t) = G_m(-\{b\}, -\{a\}, t), \tag{2.11}$$

where we introduced the short-hand notations

$$1 + \{a\} \equiv \begin{cases} \{a_1 + 1, \ldots, a_m + 1\} & \text{if } a_m < L \\ \{1, a_1 + 1, \ldots, a_{m-1} + 1\} & \text{if } a_m = L \end{cases}$$

$$-\{a\} \equiv \{L - a_m, \ldots, L - a_1\},$$

and similarly for $\{b\}$.

In this work we present two different methods to compute the propagator. The first one (to be presented in the next Section) is based on the standard spectral representation, and it uses ideas and results of the papers [6,7] which considered dynamical correlation functions in equilibrium. The second method is completely new and it is built on the Quantum Transfer Matrix approach [61]; this is presented in Section 4

## 3   The spectral representation for the propagator

Let us denote by $|\Psi_j\rangle$ a complete set of eigenstates of the Hamiltonian. The propagator can be expressed as

$$G_m(\{b\}, \{a\}, t) = \sum_{j=1}^{2^L} \frac{\langle b_1, \ldots, b_m | \Psi_j \rangle \langle \Psi_j | a_1, \ldots, a_m \rangle}{\langle \Psi_j | \Psi_j \rangle} e^{-iE_j t}, \tag{3.1}$$

where $E_j$ are the energy eigenvalues of the Hamiltonian.

The XXZ model is well known to be exactly solvable by the different versions of the Bethe Ansatz [2], which produces the eigenstates (also called Bethe states). Due to spin-$z$ conservation it is enough to consider the $\binom{L}{m}$ eigenstates in the sector with $m$ down spins. The states are characterized by a set of rapidities (also called Bethe roots) $\{\lambda_1, \ldots, \lambda_m\}$, that describe the pseudo-momenta of the interacting spin waves.

The representation (3.1) can serve as a starting point to derive exact results for the propagator. The standard idea is to transform the summation over the Bethe states into multiple integral formulas, by using the Gaudin determinant as the multi-dimensional residue for the contour integrals around a particular set of Bethe roots. However, there are several difficulties with this approach. First, one needs to know all possible positions of the Bethe roots, and then to combine and/or transform the resulting integrals into some manageable form. Second, one needs to prove the completeness of the Bethe Ansatz, which is a notoriously difficult problem, especially for the homogeneous chain. Typically the spin chain has a number of singular solutions, which need to be taken into account [62, 63].

These problems can be circumvented by a method developed earlier in the literature, namely by introducing a twist along the chain. It was proven in [6, 7] that for the dynamical correlation functions in the ground state this method yields the desired multiple integral formulas. The propagator is a closely related object, and in the following subsections we show that the methods of [6, 7] can be applied in this case too.

## 3.1 The Algebraic Bethe Ansatz

Here we briefly introduce the Algebraic Bethe Ansatz, which is the adequate framework to treat the present problem.

Let us consider the Hilbert space of the chain and also an auxiliary space $\mathbb{C}^2$ denoted with the index 0. We construct the monodromy matrix of the spin chain as

$$T(u) = R_{10}(u) \ldots R_{L0}(u) \equiv \begin{pmatrix} A_L(u) & B_L(u) \\ C_L(u) & D_L(u) \end{pmatrix}. \tag{3.2}$$

The transfer matrix is given by

$$\tau(u) = \mathrm{Tr}_0\, T(u). \tag{3.3}$$

Here $R_{jk}(u)$ is the so-called $R$-matrix acting on the spaces $j$ and $k$. We use the normalization

$$R(u) = \begin{pmatrix} 1 & 0 & 0 & 0 \\ 0 & b(u) & c(u) & 0 \\ 0 & c(u) & b(u) & 0 \\ 0 & 0 & 0 & 1 \end{pmatrix}, \tag{3.4}$$

where

$$b(u) \equiv \frac{\sinh(u)}{\sinh(u+\eta)} = P^{-1}(u+\eta/2), \tag{3.5}$$

$$c(u) = \frac{\sinh(\eta)}{\sinh(u+\eta)}. \tag{3.6}$$

The function $b$ satisfies the relation $b^{-1}(u) = b(-u-\eta)$ which will be used often in this work.

The $R$-matrix satisfies the Yang–Baxter (YB) equation

$$R_{12}(u_1-u_2)R_{13}(u_1-u_3)R_{23}(u_2-u_3) = R_{23}(u_2-u_3)R_{13}(u_1-u_3)R_{12}(u_1-u_2) \tag{3.7}$$

and the unitarity and crossing relations

$$R(u)R(-u) = 1, \qquad \frac{\sinh(u-\eta)}{\sinh(u)}R^{-1}(u) = \sigma_1^y R^{t_1}(u-\eta)\sigma_1^y, \qquad (3.8)$$

where $t_1$ denotes transposition in the first vector space. Further, the $R$-matrix satisfies the initial condition $\hat{R}(0) = P$, where $P$ is the permutation matrix of $\mathbb{C}^2 \otimes \mathbb{C}^2$: $P(v_1 \otimes v_2) = v_2 \otimes v_1$, $v_1, v_2 \in \mathbb{C}^2$. It follows from the YB relation that the transfer matrices with different spectral parameters form a commuting family:

$$[\tau(u), \tau(v)] = 0.$$

The transfer matrix satisfies the initial condition $\tau(0) = U$, where $U$ is the shift operator by one site to the left (towards decreasing lattice site indices). Also, it generates the Hamiltonian (2.1) through the relation

$$H = 2\sinh(\eta)\left.\frac{d}{du}\log\tau(u)\right|_{u=0}. \qquad (3.9)$$

As we described above, the direct evaluation of the spectral sum (3.1) would pose certain technical difficulties, which can be avoided if we first study the diagonalization of a deformed Hamiltonian [6, 7]. The idea is to introduce a twist along the chain, which will simplify the spectral representation in a certain non-physical limit. In the present work we choose to use a homogeneous twist, leading to a homogeneous Hamiltonian.

Let $\kappa \in \mathbb{C}$, $\kappa \neq 0$ be the twist parameter and let us define the twist matrix

$$M = \begin{pmatrix} 1 & 0 \\ 0 & \kappa \end{pmatrix}. \qquad (3.10)$$

The case of $\kappa = 1$ corresponds to the undeformed case.

We define the twisted monodromy matrix as

$$T^{(\kappa)}(u) = M_0 R_{10}(u)\ldots M_0 R_{L0}(u) \equiv \begin{pmatrix} A^{(\kappa)}(u) & B^{(\kappa)}(u) \\ C^{(\kappa)}(u) & D^{(\kappa)}(u) \end{pmatrix}. \qquad (3.11)$$

The twisted transfer matrix is then given by

$$\tau^{(\kappa)}(u) = \text{Tr}_0\, T^{(\kappa)}(u). \qquad (3.12)$$

The twist matrix commutes with the action of the local $R$-matrices for any two spaces with indices $a, b$:

$$M_a M_b R_{ab}(u) = R_{ab}(u)M_a M_b. \qquad (3.13)$$

Using this relation and (3.7) it can be shown that the twisted transfer matrices also form a commuting family:

$$[\tau^{(\kappa)}(u), \tau^{(\kappa)}(v)] = 0. \qquad (3.14)$$

It is important that the transfer matrices with different $\kappa$ parameters do not commute with each other.

It follows from (3.14) that the Hamiltonian defined as

$$H^{(\kappa)} = 2\sinh(\eta)\left.\frac{d}{du}\log\tau^{(\kappa)}(u)\right|_{u=0} \qquad (3.15)$$

also commutes with the transfer matrices. A direct computation gives

$$H^{(\kappa)} = \sum_{j=1}^{L}\left[\frac{2}{\kappa}\sigma_j^+\sigma_{j+1}^- + 2\kappa\sigma_j^-\sigma_{j+1}^+ + \Delta\left(\sigma_j^z\sigma_{j+1}^z - 1\right)\right], \qquad (3.16)$$

and periodic boundary conditions are understood.

This Hamiltonian is Hermitian if $|\kappa| = 1$, and we call this case the unitary twist. Writing $\kappa = e^{i\phi}$ with $\phi \in [0, 2\pi)$ the Hamiltonian can be expressed as

$$H^{(\kappa)} = H_0 + H_{DW},$$

$$H_0 = \cos(\phi) \sum_{j=1}^{L} \left[ \sigma_j^x \sigma_{j+1}^x + \sigma_j^y \sigma_{j+1}^y + \frac{\Delta}{\cos(\phi)} \left( \sigma_j^z \sigma_{j+1}^z - 1 \right) \right],$$

$$H_{DW} = -\sin(\phi) \sum_{j=1}^{L} \left[ \sigma_j^x \sigma_{j+1}^y - \sigma_j^y \sigma_{j+1}^x \right].$$

(3.17)

This Hamiltonian can be understood as an XXZ model (with a different anisotropy parameter) perturbed by a Dzyaloshinskii–Moriya interaction term [64]. This term breaks the space and spin reflection symmetries of the model, whereas the combination of these two symmetries is still preserved.

The finite dimensional matrices involved above are all analytic functions of $\kappa$, therefore the propagator can be computed as the limit

$$G_m(\{b\}, \{a\}, t) = \lim_{\kappa \to 1} G_m^{(\kappa)}(\{b\}, \{a\}, t),$$

$$G_m^{(\kappa)}(\{b\}, \{a\}, t) \equiv \langle b_1, \ldots, b_m | e^{-iH^{(\kappa)}t} | a_1, \ldots, a_m \rangle.$$

(3.18)

Note that for this limit we do not require to have a unitary twist. The hermiticity is only required if one is interested in the physical applications of $H^{(\kappa)}$, but the propagator is a well defined finite dimensional object for every $\kappa \neq 0$.

It is also important that generally the twisted Hamiltonian is not symmetric anymore, therefore the corresponding propagator loses its symmetry (2.10).

Denoting by $\left| \Psi_j^{(\kappa)} \right\rangle$ a complete set of states the twisted propagator can be expressed as

$$G_m^{(\kappa)}(\{b\}, \{a\}, t) = \sum_{j=1}^{\binom{L}{m}} \frac{\langle b_1, \ldots, b_m | \Psi_j^{(\kappa)} \rangle \langle \Psi_j^{(\kappa)} | a_1, \ldots, a_m \rangle}{\langle \Psi_j^{(\kappa)} | \Psi_j^{(\kappa)} \rangle} e^{-iE_j^{(\kappa)}t},$$

(3.19)

where $E_j^{(\kappa)}$ are the eigenvalues of the twisted Hamiltonian.

In the Algebraic Bethe Ansatz the Bethe vectors and dual vectors are defined for arbitrary sets of rapidities as

$$|\{\lambda\}_m\rangle = \prod_{j=1}^{m} B^{(\kappa)}(\lambda_j - \eta/2)|0\rangle, \qquad \langle\{\lambda\}_m| = \langle 0| \prod_{j=1}^{m} C^{(\kappa)}(\lambda_j - \eta/2).$$

(3.20)

Here the shift of $-\eta/2$ is introduced for later convenience.

The coordinate Bethe Ansatz representation of these vectors for $x_1 < \cdots < x_m$ is [2, 65]

$$\langle x_1, \ldots, x_m | \prod_{j=1}^{m} B^{(\kappa)}(\lambda_j - \eta/2)|0\rangle = \sum_{P \in \sigma_M} \prod_j F^{(\kappa)}(\lambda_{P_j}, L - x_j) \prod_{j>k} b^{-1}(\lambda_{P_k} - \lambda_{P_j})$$

$$\langle 0| \prod_{j=1}^{m} C^{(\kappa)}(\lambda_j - \eta/2)|x_1, \ldots, x_m\rangle = \sum_{P \in \sigma_M} \prod_j (\kappa F^{(\kappa)}(\lambda_{P_j}, x_j - 1)) \prod_{j<k} b^{-1}(\lambda_{P_k} - \lambda_{P_j}),$$

(3.21)

with

$$F^{(\kappa)}(\lambda, x) = \frac{\sinh(\eta)}{\sinh(\lambda + \eta/2)} (\kappa P(-\lambda))^x,$$

(3.22)

where we also defined

$$P(\lambda) \equiv e^{ip(\lambda)} = \frac{\sinh(\lambda + \eta/2)}{\sinh(\lambda - \eta/2)}. \tag{3.23}$$

These are exact formulas valid for arbitrary sets of rapidities avoiding the singular points $\pm\eta/2$.

The vectors (3.20) are right and left eigenvectors of the transfer matrices if the Bethe rapidities satisfy the Bethe equations

$$Y^{(\kappa)}(\lambda_j|\{\lambda\}) = 0, \quad j = 1\dots N, \tag{3.24}$$

where

$$Y^{(\kappa)}(\nu|\{\lambda\}) = \sinh^L(\nu+\eta/2)\prod_{j=1}^m \sinh(\lambda_j-\nu+\eta)+\kappa^L \sinh^L(\nu-\eta/2)\prod_{j=1}^m \sinh(\lambda_j-\nu-\eta). \tag{3.25}$$

Note that our $Y^{(\kappa)}$ functions differ from those of [6,7] by a simple shift, which was introduced in (3.20). Also, our $Y^{(\kappa)}$ involves the coefficient $\kappa^L$ (as opposed to simply $\kappa$), which is a result of our homogeneous twist applied at each site.

For these on-shell states the twisted transfer matrix eigenvalues are

$$\tau^{(\kappa)}(\nu|\{\lambda\}) = \prod_{j=1}^m \frac{\sinh(\lambda_j - \nu + \eta/2)}{\sinh(\lambda_j - \nu - \eta/2)} + \kappa^L (b(\nu))^L \prod_{j=1}^m \frac{\sinh(\lambda_j - \nu - 3\eta/2)}{\sinh(\lambda_j - \nu - \eta/2)}. \tag{3.26}$$

From (3.15) follows that the energy eigenvalues are

$$E = \sum_{j=1}^m \varepsilon(\lambda_j), \qquad \varepsilon(\lambda) = \frac{2\sinh^2(\eta)}{\sinh(\lambda - \eta/2)\sinh(\lambda + \eta/2)}. \tag{3.27}$$

Note that the twist $\kappa$ only enters through the Bethe equations, but the functional form of the energy is the same for all $\kappa$.

In the case of a unitary twist the left- and right eigenvectors are adjoints of each other, and this can be seen directly on the coordinate space representations. However, this is not true anymore for a generic twist $|\kappa| \neq 1$.

We remark that the Bethe equations in the original form (3.24) are completely free of singularities, and these are the equations which follow from the Algebraic Bethe Ansatz built on Lax operators with a non-singular normalization. It was emphasized for example in the work [66] by Baxter that the completeness of the Bethe Ansatz should always be investigated using these singularity-free equations.

We call a solution of the Bethe equations (3.24) admissible, if

$$\sinh^L(\lambda_k - \eta/2)\prod_{j=1}^m \sinh(\lambda_j - \lambda_k - \eta) \neq 0, \qquad k = 1\dots N. \tag{3.28}$$

A solution is called off-diagonal, if all Bethe rapidities are distinct.

For sets of rapidities avoiding the singular points $\pm\eta/2$ let us define the functions $Q_j(\{\lambda\})$ as

$$e^{Q_j(\{\lambda\})} \equiv P^L(\lambda_j) \prod_{k,k\neq j} S(\lambda_j - \lambda_k), \tag{3.29}$$

where we defined

$$S(\lambda) \equiv \frac{b(\lambda)}{b(-\lambda)} = \frac{\sinh(\lambda - \eta)}{\sinh(\lambda + \eta)}. \tag{3.30}$$

For admissible solutions the Bethe equations can be written as

$$e^{Q_j(\{\lambda\})} = \kappa^L, \qquad j = 1, \ldots, m. \tag{3.31}$$

In these conventions (and for $|\kappa| = 1$) the one particle solutions for $\Delta > 1$, $\eta \in \mathbb{R}$ are purely imaginary, whereas for $\Delta < 1$, $\eta \in i\mathbb{R}$ they are purely real.

The norm is defined as the scalar product of an eigenstate and a dual state, and it is given by [67, 68]

$$\langle 0| \prod_{j=1}^{m} C^{(\kappa)}(\lambda_j - \eta/2) \prod_{j=1}^{m} B^{(\kappa)}(\lambda_j - \eta/2)|0\rangle =$$
$$= \sinh^m(\eta) \prod_{j<k} b^{-1}(\lambda_j - \lambda_k) b^{-1}(\lambda_k - \lambda_j) \times \det \mathcal{G}, \tag{3.32}$$

where $\mathcal{G}$ is the so-called Gaudin matrix:

$$\mathcal{G}_{jk} = \frac{\partial Q_j}{\partial \lambda_k} = \delta_{j,k} \left( Lq(\lambda_j) + \sum_l \varphi(\lambda_j - \lambda_l) \right) - \varphi(\lambda_j - \lambda_k), \tag{3.33}$$

where $Q_j$ are the logarithms of the Bethe equations defined in (3.31) and

$$q(\lambda) = \frac{d}{d\lambda} \log(P(\lambda)) = -\frac{\sinh(\eta)}{\sinh(\lambda + \eta/2)\sinh(\lambda - \eta/2)} = -\frac{\varepsilon(\lambda)}{2\sinh(\eta)}, \tag{3.34}$$

$$\varphi(\lambda) = \frac{d}{d\lambda} \log(S(\lambda)) = \frac{\sinh(2\eta)}{\sinh(\lambda + \eta)\sinh(\lambda - \eta)}. \tag{3.35}$$

In order to compute the propagator we need to treat the object

$$\frac{\langle b_1, \ldots, b_m| \prod_{j=1}^{m} B^{(\kappa)}(\lambda_j - \eta/2)|0\rangle \langle 0| \prod_{j=1}^{m} C^{(\kappa)}(\lambda_j - \eta/2)|a_1, \ldots, a_m\rangle}{\langle 0| \prod_{j=1}^{m} C^{(\kappa)}(\lambda_j - \eta/2) \prod_{j=1}^{m} B^{(\kappa)}(\lambda_j - \eta/2)|0\rangle}. \tag{3.36}$$

For on-shell Bethe states this can be written as

$$\prod_{j=1}^{m} (-q(\lambda_j)) \times \frac{W^{(\kappa)}_{\{b\},\{a\}}(\{\lambda\})}{\det \mathcal{G}}, \tag{3.37}$$

where the amplitude $W^{(\kappa)}_{\{b\},\{a\}}(\{\lambda\})$ arises simply from the product of a Bethe wavefunction and a dual function, cancelling certain factors coming from the norm (3.32):

$$W^{(\kappa)}_{\{b\},\{a\}}(\{\lambda\}) = \sum_{P \in \sigma_M} \prod_j (\kappa P(-\lambda_{P_j}))^{b_j - L} \prod_{\substack{j>k \\ P_j < P_k}} S(\lambda_k - \lambda_j) \times$$
$$\times \sum_{P \in \sigma_M} \prod_j (\kappa P(-\lambda_{P_j}))^{-a_j} \prod_{\substack{j>k \\ P_j < P_k}} S(\lambda_j - \lambda_k). \tag{3.38}$$

It follows from the overall periodicity of the wave function (or from the product of the Bethe equations) that

$$\prod_j \kappa^L P^L(-\lambda_j) = 1, \tag{3.39}$$

therefore we also have

$$
W^{(\kappa)}_{\{b\},\{a\}}(\{\lambda\}) = \sum_{P\in\sigma_M} \prod_j (\kappa P(-\lambda_{P_j}))^{b_j} \prod_{\substack{j>k \\ P_j<P_k}} S(\lambda_k - \lambda_j) \times
$$
$$
\times \sum_{P\in\sigma_M} \prod_j (\kappa P(-\lambda_{P_j}))^{-a_j} \prod_{\substack{j>k \\ P_j<P_k}} S(\lambda_j - \lambda_k). \tag{3.40}
$$

It is proven in Appendix A of [7] that the Bethe vectors corresponding to the admissible off-diagonal solutions form a basis in the $N$-particle subsector, if $\kappa$ is within a punctured neighbourhood of the origin: $0 < |\kappa| < \kappa_0$, where $\kappa_0$ depends on $N$ and $L$. Therefore the spectral sum can be expressed as a sum over contour integrals encircling the Bethe roots, for details see [7] and our Appendix A.

The $\kappa$-deformed propagator can thus be expressed as

$$
G^{(\kappa)}_m(\{b\},\{a\},t) = \sum_{n=1}^{\binom{L}{m}} \prod_{j=1}^m \oint_{\mathcal{C}_j} \frac{du_j}{2\pi i} q(u_j) \times \frac{W^{(\kappa)}_{\{b\},\{a\}}(\{u\}) e^{-i(\sum_{k=1}^m \varepsilon(u_k))t}}{\prod_{k=1}^m (\kappa^L e^{Q_k(\{u\})} - 1)}, \tag{3.41}
$$

where for each term in the sum the contours $\mathcal{C}_j$ are small circles around the corresponding Bethe root $\lambda_j$ within the set $\{\lambda\}$.

The next step is to transform the sum over the small contour integrals around the sets of Bethe roots into a single common contour which surrounds the remaining singularities of the integrand. There are singular points corresponding to the diagonal solutions of the Bethe equations, but they give zero contribution due to the vanishing of the Bethe wave functions. Therefore, the remaining singularities of the integrand are only at the special points $u = \pm\eta/2$.

In complete analogy with Lemma 4.1 of [7] we construct contours

$$
\mathcal{C}_\pm(R_j) = \mathcal{C}(\eta/2, R_j) \cup \mathcal{C}(-\eta/2, R_j), \tag{3.42}
$$

where $R_j \in \mathbb{R}^+$ stands for the radius of the contours around the singular points. All the remaining singularities of the integrand are inside $\mathcal{C}_\pm(R_j)$ for $R_j$ small enough. It is important that for the multiple integrals the radiuses $R_j$ have to be chosen to be non-coinciding, otherwise we would hit singularities at $u_j - u_k = \pm\eta$, leading to ill-defined integrals.

This way we obtain

$$
G^{(\kappa)}_m(\{b\},\{a\},t) = \frac{1}{m!} \prod_{j=1}^m \oint_{\mathcal{C}_\pm(R_j)} \frac{du_j}{2\pi i} q(u_j) \times \frac{W^{(\kappa)}_{\{b\},\{a\}}(\{u\}) e^{-i(\sum_{k=1}^m \varepsilon(u_k))t}}{\prod_{k=1}^m (1 - \kappa^L e^{Q_k(\{u\})})}. \tag{3.43}
$$

Here the factor $1/m!$ was introduced to cancel the permutation symmetry of the integrals.

We can simplify the wave function amplitude: If we expand one sum over permutations, and perform an exchange of the integration variables in each term separately, the arising $S$-factors can be compensated and we obtain

$$
G^{(\kappa)}_m(\{b\},\{a\},t) = \frac{1}{m!} \prod_{j=1}^m \oint_{\mathcal{C}_\pm(R_j)} \frac{du_j}{2\pi i} q(u_j) \times \frac{\Psi^{(\kappa)}_{\{b\},\{a\}}(\{u\}) e^{-i(\sum_{k=1}^m \varepsilon(u_k))t}}{\prod_{k=1}^m (1 - \kappa^L e^{Q_k(\{u\})})}, \tag{3.44}
$$

where for the amplitude we can use two alternative forms:

$$
\Psi^{(\kappa)}_{\{b\},\{a\}}(\{u\}) = \prod_j (\kappa^{-1} P(u_j))^{a_j} \sum_{P\in\sigma_M} \prod_j (\kappa^{-1} P(u_{P_j}))^{-b_j} \prod_{\substack{j>k \\ P_j<P_k}} S(u_k - u_j) \tag{3.45}
$$

and

$$\Psi^{(\kappa)}_{\{b\},\{a\}}(\{u\}) = \prod_j (\kappa^{-1}P(u_j))^{-b_j} \sum_{P\in\sigma_M} \prod_j (\kappa^{-1}P(u_{P_j}))^{a_j} \prod_{\substack{j>k \\ P_j<P_k}} S(u_j - u_k).$$ (3.46)

This idea to express the product of two sums over permutations by a simple sum over permutations was also used in the earlier work on infinite volume propagators, see [60] and [53–56].

Note that if we set $\kappa = 1$, then the two different forms (3.45)-(3.46) reflect the symmetry (2.10) of the propagator.

## 3.2 Continuing back to the untwisted case

The physical propagator is obtained by analytically continuing the formula (3.44) in $\kappa$, from a neighbourhood of zero to $\kappa = 1$. The integrand is an analytic function of $\kappa$, and in fact $\kappa$ only enters in the denominator. Therefore, the only non-analyticity that can happen during the procedure is when some singularities of this denominator cross the contours. Such singularities occur at the solution of the Bethe equations. A multi-dimensional pole could be picked up when for a given solution all Bethe rapidities would cross the contours.

It is known that there are singular solutions of the untwisted model that include the rapidities $\pm\eta/2$ [62] and they have been studied by essentially the same twisting procedure in [63]. The available data from concrete examples in small volumes show that as the twist parameter $\kappa$ is continued back to 1, only those types of singular states are produced which include these singular rapidities with multiplicity one at most. Therefore the only case when the analytic continuation of the integrals can produce extra contributions is for $N = 2$, when the two Bethe rapidities approach $\pm\eta/2$. It is shown in Appendix A that even in these cases there is no addtional pole contribution.

We thus obtain our final result for the untwisted case:

$$G_m(\{b\},\{a\},t) = \prod_{j=1}^m \left( \oint_{\mathcal{C}_\pm(R_j)} \frac{du_j}{2\pi i} q(u_j) \right) \times \frac{\Psi_{\{b\},\{a\}}(\{u\}_m)}{\prod_{k=1}^m \left(1 - e^{Q_k(\{u\})}\right)} e^{-i\left(\sum_{k=1}^m \varepsilon(u_k)\right)t},$$ (3.47)

where $\Psi_{\{b\},\{a\}}(\{u\}_m) = \Psi^{(1)}_{\{b\},\{a\}}(\{u\}_m)$. We stress once more that the integrals are well defined (and numerically stable) only if the radiuses $R_j$ are non-coinciding.

# 4 The propagator from the Trotter decomposition

As an alternative to the previous method we also compute the propagator by a lattice path integral, in close analogy with the study of the thermodynamical state functions of the model [61]. In this Section we work mostly with the untwisted model, because this technique does not require the introduction of the twist parameter $\kappa$.

For future use we introduce the space reflected transfer matrix, which is defined as

$$\tilde{\tau}(u) = \mathrm{Tr}_0 \, \tilde{T}(u), \qquad \tilde{T}(u) = R_{L0}(u)\dots R_{10}(u).$$ (4.1)

It satisfies the initial condition $\tilde{\tau}(0) = U^{-1}$, and it also generates the Hamiltonian by the same relation as (3.9).

It follows that the two transfer matrices $\tau(u)$ and $\tilde{\tau}(u)$ can be used to generate the time evolution operator through a Trotter approximation. For any $s \in \mathbb{C}$

$$e^{-sH} = \lim_{N\to\infty} \left(1 - \frac{sH}{N}\right)^N = \lim_{N\to\infty} \left(\tau(-\beta/2N)\tilde{\tau}(-\beta/2N)\right)^N,$$ (4.2)

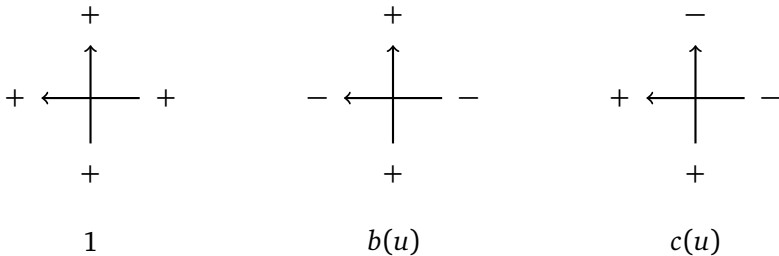

Figure 1: The vertex weights of the six vertex model. The functions $b(u)$ and $c(u)$ are given by (3.5) and (3.6), respectively. The $R$-matrix acts from bottom to top and right to left.

where

$$\beta = 2\sinh(\eta)s. \tag{4.3}$$

For our purposes it is convenient to use the crossing relation (3.8) to relate the two transfer matrices to each other:

$$\tilde{\tau}(u) = \left(\frac{\sinh(u)}{\sinh(u+\eta)}\right)^{L} \tau(-u-\eta). \tag{4.4}$$

This leads to the following form of the Trotter decomposition:

$$e^{-sH} = \lim_{N\to\infty}\left(\left(\frac{\sinh(-\beta/(2N))}{\sinh(-\beta/(2N)+\eta)}\right)^{L} \tau(-\beta/(2N))\tau(\beta/(2N)-\eta)\right)^{N}. \tag{4.5}$$

The advantage of this representation is that it only uses the same transfer matrix.

For technical reasons it is better to use a set of non-coinciding inhomogeneities. Therefore we define $\beta_j$ with $j = 1,\ldots,N$ such that $|\beta_j - \beta_k| = \mathcal{O}(1/N)$. Focusing on real times we thus write

$$e^{-itH} = \lim_{N\to\infty}\prod_{j=1}^{N}\left(\left(\frac{\sinh\big(-i\beta_j/(2N)\big)}{\sinh\big(-i\beta_j/(2N)+\eta\big)}\right)^{L} \tau\big(-i\beta_j/(2N)\big)\tau\big(i\beta_j/(2N)-\eta\big)\right), \tag{4.6}$$

with

$$\beta_j = 2\sinh(\eta)t + \mathcal{O}(1/N). \tag{4.7}$$

The propagator is thus expressed as

$$\begin{aligned}
G_m(\{b\},\{a\},t) = \lim_{N\to\infty}\Bigg(&\prod_{j=1}^{N}\left(\frac{\sinh\big(-i\beta_j/(2N)\big)}{\sinh\big(-i\beta_j/(2N)+\eta\big)}\right)^{L} \times\\
&\times\langle b_1,\ldots,b_m|\prod_{j=1}^{N}\tau\big(-i\beta_j/(2N)\big)\tau\big(i\beta_j/(2N)-\eta\big)|a_1,\ldots a_m\rangle\Bigg).
\end{aligned} \tag{4.8}$$

Due to the correspondence between the XXZ model and the six vertex model, at any finite $N$ the above expression is equal to a six vertex partition function of an $L \times 2N$ square lattice, with particular boundary conditions. The individual weights of the six vertex model are depicted on Fig. 1, whereas an example for the partition function is shown in Fig. 2. Here the presence of the transfer matrices leads to periodic boundary conditions in the space direction (chosen as the horizontal direction), whereas in the time direction (chosen as the vertical direction) we have fixed boundary conditions given by the initial and final states $|a_1,\ldots,a_m\rangle$ and $\langle b_1,\ldots,b_m|$.

The six vertex model is invariant under a reflection along the North-West diagonal. Due to this reflection symmetry of the $R$-matrix, any such partition function can be evaluated alternatively in the so-called quantum channel (also called the rotated or mirror channel). We can thus build an alternative monodromy matrix that acts on an inhomogeneous spin chain of length $2N$, such that the inhomogeneities are determined by the spectral parameters of the transfer matrices in (4.8). To be precise we define

$$T^{QTM}(u) = R_{2N,0}\left(u - \frac{i\beta}{2N}\right)R_{2N-1,0}\left(u + \frac{i\beta}{2N} - \eta\right)\dots R_{20}\left(u - \frac{i\beta}{2N}\right)R_{10}\left(u + \frac{i\beta}{2N} - \eta\right) =$$
$$= \begin{pmatrix} A(u) & B(u) \\ C(u) & D(u) \end{pmatrix}. \tag{4.9}$$

The initial and final states of the propagator become boundary conditions in the quantum channel, and (due to the periodic boundary conditions in the spatial direction) the partition function can be evaluated as a trace of a particular ordered product of the monodromy matrix elements $A(0), B(0), C(0)$ or $D(0)$. The explicit relation is

$$\langle b_1, \dots, b_m | \prod_{j=1}^{N} \tau\left(-i\beta_j/(2N)\right)\tau\left(i\beta_j/(2N) - \eta\right)|a_1, \dots a_m\rangle = \mathrm{Tr}\prod_{j=1}^{L} T^{QTM}_{s_j^b, s_j^a}(0), \tag{4.10}$$

where $s_j^{a,b}$ are the the spin components at position $j$ in the initial and final states, respectively. Explicitly

$$s_j^a = \begin{cases} 1 & \text{if} \quad j \in \{a_1, \dots, a_m\} \\ 2 & \text{if} \quad j \notin \{a_1, \dots, a_m\}, \end{cases} \tag{4.11}$$

and similarly for $s_j^b$.

The normalized expression for the propagator is then

$$G_m(\{b\}, \{a\}, t) = \lim_{N\to\infty}\left[\prod_{j=1}^{N}\left(\frac{\sinh\left(-i\beta_j/(2N)\right)}{\sinh\left(-i\beta_j/(2N) + \eta\right)}\right)^L \times \mathrm{Tr}\prod_{j=1}^{L} T^{QTM}_{s_j^b, s_j^a}(0)\right]. \tag{4.12}$$

The propagator enjoys a complete spin flip invariance, and as a result an equivalent expression is

$$G_m(\{b\}, \{a\}, t) = \lim_{N\to\infty}\left[\prod_{j=1}^{N}\left(\frac{\sinh\left(-i\beta_j/(2N)\right)}{\sinh\left(-i\beta_j/(2N) + \eta\right)}\right)^L \times \mathrm{Tr}\prod_{j=1}^{L} T^{QTM}_{\tilde{s}_j^b, \tilde{s}_j^a}(0)\right], \tag{4.13}$$

where $\tilde{s}_j^{a,b} = 3 - s_j^{a,b}$ or explicitly

$$s_j^a = \begin{cases} 2 & \text{if} \quad j \in \{a_1, \dots, a_m\} \\ 1 & \text{if} \quad j \notin \{a_1, \dots, a_m\}, \end{cases} \tag{4.14}$$

and similarly for $s_j^b$. In this work we will use the representation (4.13) because this leads conforms to certain conventions used in the construction of the so-called F-basis, to be presented below.

As an example for these formulas, we consider $L = 6$ and a specific matrix element of the two particle propagator:

$$\langle 1, 3 | e^{-iHt} | 2, 3 \rangle. \tag{4.15}$$

According to (4.10), this is proportional to $\mathrm{Tr}\left[B(0)C(0)A(0)D(0)D(0)D(0)\right]$.

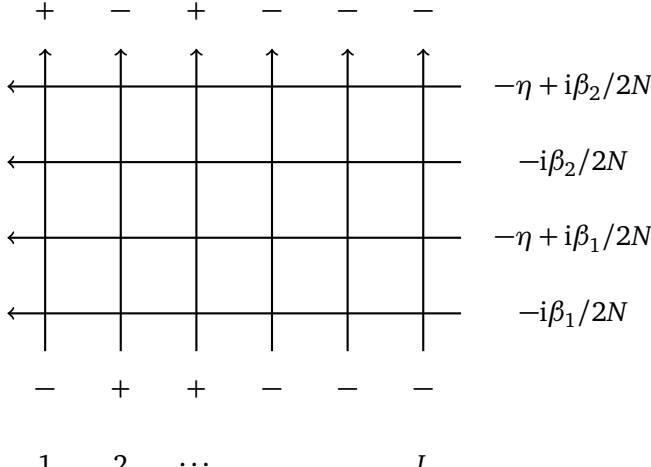

Figure 2: An example for the partition function (4.10) with $L = 6$ and $N = 2$, describing the matrix element (4.15). We have periodic boundary conditions in the horizontal direction, and the bottom and top rows are fixed by the initial and final states in the spin basis.

We note that the ordering of the inhomogeneities in (4.9) does not influence the traces that we intend to compute. On the one hand, this follows from the commutativity of the transfer matrices $\tau$ in (4.5). On the other hand, this symmetry will be explicit after the introduction of the F-basis in Section 4.1.

The symmetry (2.10) of the propagator can be observed at finite Trotter number as well. Starting from the expression (4.9) for the quantum monodromy matrix we can perform a series of crossing transformations on the $R$-matrices leading to

$$T^{QTM}(0) = S \times (\tilde{T}_{QTM}(0))^{t_0} \times S, \tag{4.16}$$

where

$$\tilde{T}_{QTM}(0) = R_{2N\,0}\left(\tfrac{i\beta}{2N} - \eta\right) R_{2N-1\,0}\left(-\tfrac{i\beta}{2N}\right) \ldots R_{20}\left(\tfrac{i\beta}{2N} - \eta\right) R_{10}\left(-\tfrac{i\beta}{2N}\right) \tag{4.17}$$

and

$$S = \prod_{j=1}^{2N} \sigma_j^y. \tag{4.18}$$

Note that in (4.17) the order of the inhomogeneities has been modified as a result of the crossing, but this does not effect the traces. Also, the action of the $S$ operators also drops out due to $S^2 = 1$. Thus it follows from (4.16) that the initial and final states can be exchanged, and an alternative formula for (4.13) is

$$G_m(\{b\}, \{a\}, t) = \lim_{N \to \infty} \left[ \prod_{j=1}^{N} \left( \frac{\sinh\left(-i\beta_j/(2N)\right)}{\sinh\left(-i\beta_j/(2N) + \eta\right)} \right)^L \times \mathrm{Tr} \prod_{j=1}^{L} T_{\tilde{s}_j^a, \tilde{s}_j^b}^{QTM}(0) \right]. \tag{4.19}$$

Our goal is to compute the traces (4.13)-(4.19) at finite $N$ using exact methods, and to take the $N \to \infty$ limit afterwards. We will start with low particle numbers, but we will consider arbitrary $L$ volumes. Typically there will be a large number of $D$ operators in the product (corresponding to the vacuum with the down spins), and a smaller number of $B$, $C$, and $A$ operators depending on the initial and final positions of the excitations.

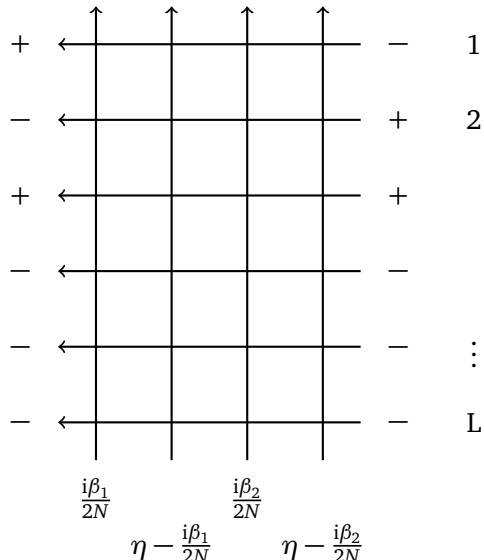

Figure 3: An example for the partition function (4.10) after the reflection along the North-West diagonal. The horizontal lines are now interpreted as the matrix elements of $T_{QTM}$. We have periodic boundary conditions in the vertical direction, which amounts to taking the trace of a certain product of $A, B, C, D$ operators. In the present case we have $\mathrm{Tr}\, BCAD^{L-2}$.

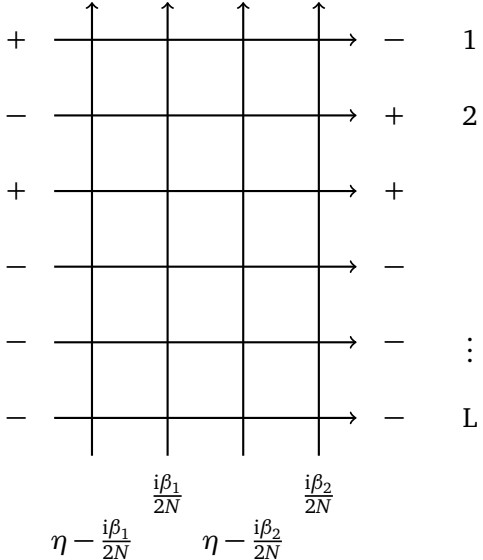

Figure 4: An example for the partition function (4.10) after a crossing transformation performed on the horizontal lines, which are now interpreted as the matrix elements of $\tilde{T}_{QTM}$. According to (4.16) this partition function is given by $\mathrm{Tr}\, CBAD^{L-2}$.

We stress that in the usual QTM method one deals with the powers of Tr $T^{QTM}(0) = A(0) + D(0)$, and the partition function is computed only in the $L \to \infty$ limit. This leads to the major simplification that only the leading eigenvalue of Tr $T^{QTM}$ needs to be considered. In contrast, here we are dealing with the individual matrix elements of the monodromy matrix, we keep the volume $L$ at a fixed finite value, and evaluate the product (4.13)-(4.19) exactly.

The direct evaluation of the traces of the product of operators in (4.13)-(4.19) would be quite cumbersome, due to the complicated forms of the $A, B, C$, and $D$ operators. A remarkable simplification can be achieved by performing an appropriate basis transformation. We compute the traces in the so-called $F$-basis, whose big advantage is that the $D$ operators are diagonal and the $B$, $C$, and $A$ operators take also sufficiently simple forms. This way we obtain manageable expressions for the propagator.

In the remainder of this Section we will work with (4.19) instead of (4.13). The reason for this is simply that in the conventions that we are using it leads to more transparent prescriptions for constructing the propagator.

## 4.1 The $F$-basis

The factorizing $F$-matrices of Maillet and Santos were introduced in [69] and later used in [70] for the calculation of the form factors in the spin chain. From a computational point of view, their main advantage is that in the basis generated by the $F$-matrices (the so-called $F$-basis) the monodromy matrix elements take very simple forms and their expression is completely symmetric with respect to the ordering of the quantum spaces of the chain. In the following we give a brief overview on the $F$-basis of the XXZ model. We will not repeat the derivations, rather we will just cite the formulas necessary for our computations. For a diagrammatic introduction to the $F$-matrices we refer the reader to [71], and we also note that an interesting alternative derivation is also given in [72].

With full generality we consider a spin chain of length $n$ with a set of inhomogeneities $\xi_j$, $j = 1 \ldots n$. As usually, the monodromy matrix is

$$T(u) = R_{n0}(u - \xi_n) \ldots R_{10}(u - \xi_1) = \begin{pmatrix} A(u) & B(u) \\ C(u) & D(u) \end{pmatrix},$$

$$\tau(u) = \mathrm{Tr}_0 \, T(u).$$

(4.20)

For the main goal of this paper, namely the computation of the propagator, the $F$-basis will be established in the QTM channel and we will identify $n = 2N$, and the inhomogeneities will be set to

$$\xi_j = \begin{cases} i\beta_j/(2N) & j = 1, \ldots, N \\ -i\beta_{j-N}/(2N) + \eta & j = N+1, \ldots, 2N. \end{cases}$$

(4.21)

Here we re-ordered the set of inhomogeneity parameters, but this does not effect the computation of the traces, as already remarked earlier. In the present section our goal is just to introduce the necessary formulas for the $F$-basis, therefore we keep $n$ and $\xi_j$ as arbitrary parameters.

For any permutation $\pi \in S_n$ we can uniquely define a matrix $R^\pi_{1\ldots n}(\xi_1, \ldots, \xi_n)$ that effectively permutes the fundamental vector spaces and the corresponding inhomogeneities in the construction of the transfer matrix. The main idea is that for any elementary exchange $(\xi_j, \xi_{j+1}) \to (\xi_{j+1}, \xi_j)$ we introduce the action of $\hat{R}_{j,j+1}(\xi_j - \xi_{j+1})$ acting only on the vector spaces of the sites $j$ and $j+1$. The total $R^\pi$ is constructed as the product of such operations as the full permutation $\pi$ is constructed using the elementary ones. The uniqueness of the construction is guaranteed by the Yang-Baxter equation. The $R^\pi$ matrices defined this way satisfy the relation

$$R^\pi_{1\ldots n}(\xi_1, \ldots, \xi_n) T_{0,1\ldots n}(u; \xi_1, \ldots \xi_n) = T_{0,\pi(1)\ldots\pi(n)}(u; \xi_{\pi(1)}, \ldots, \xi_{\pi(n)}) R^\pi_{1\ldots n}(\xi_1, \ldots, \xi_n).$$

(4.22)

The factorizing $F$-matrix is an invertible matrix $F_{1\ldots n}(\xi,\ldots,\xi_n)$ which satisfies the following condition for any $\pi \in S_n$:

$$F_{\pi(1)\ldots\pi(n)}(\xi_{\pi(1)},\ldots,\xi_{\pi(n)})R^{\pi}_{1\ldots n}(\xi_1,\ldots\xi_n) = F_{1\ldots n}(\xi_1,\ldots,\xi_n). \tag{4.23}$$

In other words it factorizes the (composite) $R$-matrix.

It was shown in [69] that the $F$-matrices can be constructed recursively, by increasing the length of the spin chain at each step. Alternatively, they can be computed by a summation over the matrices $R^{\pi}$ [72]. However, these formulas will not be needed in the present paper, therefore we refer the reader to the original papers.

The important property of the $F$-matrices that will be used in our calculations is, that the monodromy matrix elements take especially simple form in the basis generated by them. Let us perform a basis transformation in the physical space (and keep the auxiliary space unchanged), and let us denote by $\tilde{T}$ the monodromy matrix in the new basis:

$$\tilde{T}(u,;\xi_1,\ldots,\xi_n) = \begin{pmatrix} \tilde{A}(u) & \tilde{B}(u) \\ \tilde{C}(u) & \tilde{D}(u) \end{pmatrix} \equiv F_{1\ldots n}(\xi_1,\ldots,\xi_n)\begin{pmatrix} A(u) & B(u) \\ C(u) & D(u) \end{pmatrix} F^{-1}_{1\ldots n}(\xi_1,\ldots,\xi_n). \tag{4.24}$$

It follows from the relation (4.23) that the matrix representation in the new basis is completely symmetric with respect to the sites and inhomogeneities.

It was shown in [69] that the $F$-matrices given there lead to the following diagonal form for the $\tilde{D}$-operator:

$$\tilde{D}_{1\ldots n}(u;\xi_1,\ldots,\xi_n) = \otimes^n_{i=1}\begin{pmatrix} b(u,\xi_i) & 0 \\ 0 & 1 \end{pmatrix}_{[i]}. \tag{4.25}$$

Note, that this expression is completely symmetric under simultaneous permutation of the vector spaces $i$ and the corresponding spectral parameters $\xi_i$.

Analogously, the other elements of the monodromy matrix are given by

$$\tilde{B}_{1\ldots n}(u;\xi_1,\ldots,\xi_n) = \sum_{i=1}^{n} \sigma_i^- c(u,\xi_i) \otimes_{j\neq i}\begin{pmatrix} b(u,\xi_j) & 0 \\ 0 & b^{-1}(\xi_j,\xi_i) \end{pmatrix}_{[j]}$$

$$\tilde{C}_{1\ldots n}(u;\xi_1,\ldots,\xi_n) = \sum_{i=1}^{n} \sigma_i^+ c(u,\xi_i) \otimes_{j\neq i}\begin{pmatrix} b(u,\xi_j)b^{-1}(\xi_i,\xi_j) & 0 \\ 0 & 1 \end{pmatrix}_{[j]} \tag{4.26}$$

$$\tilde{A}_{1\ldots n}(u;\xi_1,\ldots,\xi_n) = \otimes^n_{i=1}\begin{pmatrix} 1 & \\ & b(u-\eta,\xi_i) \end{pmatrix}_{[i]} +$$

$$+ \sum_{i=1}^{n} c^2(u,\xi_i)b^{-1}(u,\xi_i)\sigma_i^-\sigma_i^+ \otimes_{j,j\neq i}\begin{pmatrix} b(u,\xi_j)b^{-1}(\xi_i,\xi_j) & \\ & b^{-1}(\xi_j,\xi_i) \end{pmatrix}_{[j]} +$$

$$+ \sum_{\substack{i,j \\ i\neq j}} c(u,\xi_i)b^{-1}(\xi_j,\xi_i)\sigma_i^- \otimes c(u,\xi_j)\sigma_j^+ \otimes_{k,k\neq i,j}\begin{pmatrix} b(u,\xi_k)b^{-1}(\xi_j,\xi_k) & \\ & b^{-1}(\xi_k,\xi_i) \end{pmatrix}_{[k]}. \tag{4.27}$$

Here $\sigma_i^{\pm}$ are the usual spin raising and lowering operators acting on site $i$: $\sigma^{\pm} = \frac{1}{2}(\sigma^x \pm i\sigma^y)$. Note, that while $\tilde{D}$, $\tilde{B}$, $\tilde{C}$ are formally the same as in [69], $\tilde{A}$ is formally different. The reason for this is that in [69] only the $SU(2)$-symmetric XXX$-\frac{1}{2}$ case was considered, where there are some special identities for the rational functions involved. On the other hand, the formula for $\tilde{A}$ can be computed using the quantum determinant (this was already remarked in [70] and

the computation was performed earlier by other researchers [73]) or by taking the limit of the formulas for the XYZ model published in [74]. For the sake of completeness we present the detailed derivation in the Appendix B, together with an alternative form (B.5).

The advantage of the $F$-basis is that it provides polarization-free expressions for the $B$, $C$ and $A$ operators. By this we mean that the action of the particle creation and annihilation is dressed only diagonally. In contrast, in the original spin basis we would be dealing with expressions of the type

$$B_{1...n}(u) = \sum_{i=1}^{n} \sigma_i^- \Omega_i + \sum_{\substack{i,j,k \\ i \neq j \neq k}} \sigma_i^- \sigma_j^- \sigma_k^+ \Omega_{ijk} + \text{higher terms}, \qquad (4.28)$$

where $\Omega_i$, $\Omega_{ijk}$,... are diagonal operators on all sites but on site $i$, site $i, j, k$, etc., respectively. Multiplying such sums would be a practically unfeasible task. On the other hand, the computation of the products of diagonally dressed operators is relatively straightforward.

We also remind that the $\tilde{A}, \tilde{B}, \tilde{C}, \tilde{D}$ operators satisfy the same commutation relations as $A, B, C, D$.

In the next two sections we employ the $F$-basis to compute the traces of products of monodromy matrix elements to obtain the propagator as given by (4.19).

## 4.2 The propagator: One particle case

In this section, we compute the one particle propagator. This is a very simple object, which could be calculated even without using any methods of integrability. Nevertheless we perform the detailed computations using the $F$-basis, which serves as a good warm up for the more complicated cases.

According to the rule (4.19) there are two different cases, that need to be treated separately:

- The particle moves by $\ell$ sites with $l = 1 \ldots L-1$. In this case we are dealing with a trace of the form

$$\langle k | e^{-iHt} | k - \ell \rangle \quad \sim \quad \text{Tr } D^{L-1-\ell}(0) B(0) D^{\ell-1}(0) C(0).$$

  The simplest case is when $\ell = 1$, this is treated in 4.2.1, whereas the generic case of $\ell = 2 \ldots L-1$ is considered in 4.2.2.

- The particle stays at its position. In this case we are dealing with a trace of the form

$$\langle k | e^{-iHt} | k \rangle \quad \sim \quad \text{Tr } D^{L-1}(0) A(0).$$

For the computations of the traces we will use the $F$-basis; the original spin basis of the QTM will not be used anymore. Therefore, in the rest of the paper it is understood that all $A, B, C, D$ operators are given by their concrete matrix representations in the $F$-basis. Also, all monodromy matrix elements will be evaluated at the spectral parameter $u = 0$, therefore we will drop this from the notation and understand that $A \equiv A(0)$, etc.

The computations of the traces will be performed algebraically using arbitrary inhomogeneities $\xi_j$, $j = 1 \ldots n$. We will use the following notations:

$$b_i \equiv b(-\xi_i) = \frac{\sinh(-\xi_i)}{\sinh(-\xi_i + \eta)} \qquad\qquad b_{i0}^{-1} \equiv b^{-1}(\xi_i) = \frac{\sinh(\xi_i + \eta)}{\sinh(\xi_i)}$$

$$b_{ij}^{-1} \equiv b^{-1}(\xi_i - \xi_j) = \frac{\sinh(\xi_i - \xi_j + \eta)}{\sinh(\xi_i - \xi_j)} \qquad\qquad c_i \equiv c(-\xi_i) = \frac{\sinh(\eta)}{\sinh(u - \xi_i + \eta)}.$$

When turning to the Trotter limit, we will set $n = 2N$ and the inhomogeneities will be specified according to (4.21). Finally, the $\beta_j$ parameters will be sent to $\beta = 2\sinh(\eta)t$ with $t$ being the physical time parameter.

### 4.2.1 One particle propagation by one site

In our framework the simplest case is when one particle hops one site. According to (4.19) this is evaluated as

$$\langle k|e^{-iHt}|k-1\rangle = \lim_{N\to\infty}\left[\prod_{j=1}^{N}\left(\frac{\sinh(-i\beta_j/2N)}{\sinh(-i\beta_j/2N+\eta)}\right)^L \mathrm{Tr}\left(D^{L-2}BC\right)\right]. \tag{4.29}$$

First, we compute the $D^{L-2}BC$ product. We can denote the structure of $B$ and $C$ as

$$
\begin{aligned}
B &= \sum_i \mathcal{B}_i \otimes_{j,j\neq i} \beta_j^{(i)}\,,\\
C &= \sum_i \mathcal{C}_i \otimes_{j,j\neq i} \gamma_j^{(i)}\,,
\end{aligned}
\tag{4.30}
$$

where $\mathcal{B}_i$, $\mathcal{C}_i$ are off-diagonal matrices in the $i$th space, and $\beta_j^{(i)}$, $\gamma_j^{(i)}$ are diagonal matrices in the $j$th space, depending on index $i$, through its parameters. The product of these two is:

$$BC = \sum_i \mathcal{B}_i\mathcal{C}_i \otimes_{j,j\neq i} \beta_j^{(i)}\gamma_j^{(i)} + \sum_{\substack{i,j\\i\neq j}} \mathcal{B}_i\gamma_i^{(j)} \otimes \beta_j^{(i)}\mathcal{C}_j \otimes_{k,k\neq i,j} \beta_k^{(i)}\gamma_k^{(j)}, \tag{4.31}$$

where we separated the terms depending on whether the off-diagonal matrices are on the same site (first sum) or not (second sum).
Hence the $BC$ product:

$$BC = \sum_{i=1}^{n} c_i^2 \sigma_i^- \sigma_i^+ \otimes_{j\neq i}\begin{pmatrix} b_j^2 b_{ij}^{-1} & 0 \\ 0 & b_{ji}^{-1} \end{pmatrix}_{[j]} + \sum_{i,j,i\neq j} c_i b_i b_{ji}^{-1}\sigma_i^- \otimes c_j b_j \sigma_j^+ \otimes_{\substack{k\\k\neq i,j}}\begin{pmatrix} b_k^2 b_{jk}^{-1} & 0 \\ 0 & b_{ki}^{-1} \end{pmatrix}_{[k]}. \tag{4.32}$$

As $D$ is diagonal, multiplying by $D^{L-2}$ is simple:

$$
\begin{aligned}
D^{L-2}BC &= \sum_{i=1}^{n} c_i^2 \sigma_i^- \sigma_i^+ \otimes_{j,j\neq i}\begin{pmatrix} b_j^L b_{ij}^{-1} & 0 \\ 0 & b_{ji}^{-1} \end{pmatrix}_{[j]} + \\
&\quad + \sum_{i,j,i\neq j} c_i b_i b_{ji}^{-1}\sigma_i^- \otimes c_j b_j^{L-1}\sigma_j^+ \otimes_{\substack{k\\k\neq i,j}}\begin{pmatrix} b_k^L b_{jk}^{-1} & 0 \\ 0 & b_{ki}^{-1} \end{pmatrix}_{[k]}.
\end{aligned}
\tag{4.33}
$$

Taking the trace gives

$$\mathrm{Tr}\left(D^{L-2}BC\right) = \sum_{i=1}^{n}\mathrm{Tr}\left(c_i^2\sigma_i^-\sigma_i^+\right)\prod_{j,j\neq i}\mathrm{Tr}\begin{pmatrix} b_j^L b_{ij}^{-1} & 0 \\ 0 & b_{ji}^{-1} \end{pmatrix}_{[j]} = \sum_{i=1}^{n} c_i^2 \prod_{j,j\neq i}\left(b_j^L b_{ij}^{-1} + b_{ji}^{-1}\right). \tag{4.34}$$

Note, that the second sum from (4.33) dropped out automatically due to its tracelessness.
We substitute this expression back to the propagator:

$$\langle k|e^{-iHt}|k-1\rangle \approx$$

$$\prod_{i=1}^{N}\left(\frac{\sinh\left(\frac{-i\beta_j}{2N}\right)}{\sinh\left(\frac{-i\beta_j}{2N}+\eta\right)}\right)^L \times \sum_{i=1}^{2N} c^2(-\xi_i)\prod_{j,j\neq i}\left(b^L(-\xi_j)b^{-1}(\xi_i-\xi_j)+b^{-1}(\xi_j-\xi_i)\right). \tag{4.35}$$

It is easy to see, that this expression is singular due to $b(0)=0$, and the homogeneous limit of $\xi_j \to \xi_k$ can not be taken directly. This was the main reason behind the introduction of the set

of non-coinciding $\beta_j$ parameters. Nevertheless the expression can be evaluated as a contour integral, by the use of the following identity.

Consider the meromorphic functions $f(u)$ and $g_j(u)$, and a contour $\mathcal{C}$ such that each $g_j$ has a simple pole at $z_j$ within the contour and $f$ does not have poles inside $\mathcal{C}$. Then

$$\oint_{\mathcal{C}} \frac{du}{2\pi i} f(u) \prod_j g_j(u) = \sum_{z_i} f(z_i) \text{Res}_{z=z_i} g_i(z) \prod_{j \neq i} g_j(z_i). \tag{4.36}$$

In our case the set of inhomogeneities $\xi_j$, $j = 1 \dots 2N$ is given by (4.21). It follows that the corresponding functions will have poles in the neighborhood of $z = 0$ and $z = \eta$. Therefore we define $\mathcal{C}$ to be an union of two small contours around $0$ and $\eta$: $\mathcal{C} = \mathcal{C}_0 \cup \mathcal{C}_\eta$. We apply the above identity with the choice

$$\begin{aligned}
f(u) &= \frac{c^2(-u)}{\sinh(\eta)(b^L(-u)-1)} \\
g_j(u) &= b^L(-\xi_j) b^{-1}(u-\xi_j) + b^{-1}(\xi_j - u).
\end{aligned} \tag{4.37}$$

One can easily see that $f$ is indeed free of poles in $\mathcal{C}$. This leads to the integral representation

$$\langle k | e^{-iHt} | k-1 \rangle =$$
$$= \lim_{N \to \infty} \left[ \prod_{j=1}^{N} \left( \frac{\sinh(-i\beta_j/2N)}{\sinh(-i\beta_j/2N+\eta)} \right)^L \oint_{\mathcal{C}} \frac{du}{2\pi i} \frac{\sinh(\eta)}{\sinh^2(-u+\eta)} \frac{1}{\frac{\sinh^L(u)}{\sinh^L(u-\eta)} - 1} \times \right.$$
$$\left. \times \prod_{j=1}^{2N} \frac{\sinh^L(\xi_j)}{\sinh^L(\xi_j+\eta)} \frac{\sinh(u-\xi_j+\eta)}{\sinh(u-\xi_j)} + \frac{\sinh(\xi_j-u+\eta)}{\sinh(\xi_j-u)} \right]. \tag{4.38}$$

After substituting (4.21) we are now free to take the homogeneous limit. This leads to

$$\langle k | e^{-iHt} | k-1 \rangle = \lim_{N \to \infty} \oint_{\mathcal{C}} \frac{du}{2\pi i} \frac{\sinh(\eta)}{\sinh^2(-u+\eta)} \frac{1}{\frac{\sinh^L(u)}{\sinh^L(u-\eta)} - 1} \times$$
$$\times \left( \frac{\sinh^L(-\frac{i\beta}{2N})}{\sinh^L(-\frac{i\beta}{2N}+\eta)} \frac{\sinh(u-\frac{i\beta}{2N}+\eta)}{\sinh(u-\frac{i\beta}{2N})} + \frac{\sinh(\frac{i\beta}{2N}-u+\eta)}{\sinh(\frac{i\beta}{2N}-u)} \right)^N \times$$
$$\times \left( \frac{\sinh(u+\frac{i\beta}{2N})}{\sinh(u+\frac{i\beta}{2N}-\eta)} + \frac{\sinh^L\left(\frac{-i\beta}{2N}\right)}{\sinh^L\left(\frac{-i\beta}{2N}+\eta\right)} \frac{\sinh(-\frac{i\beta}{2N}-u+2\eta)}{\sinh(-\frac{i\beta}{2N}+\eta-u)} \right)^N. \tag{4.39}$$

To take the Trotter limit, note that each term involving $\sinh^L(\beta/N)$ is sub-leading, irrespective of the value of $u$ under the integral. Therefore we get

$$\langle k | e^{-iHt} | k-1 \rangle = \lim_{N \to \infty} \oint_{\mathcal{C}} \frac{du}{2\pi i} \frac{\sinh(\eta)}{\sinh^2(-u+\eta)} \frac{1}{\frac{\sinh^L(u)}{\sinh^L(u-\eta)} - 1} \times$$
$$\times \left( \frac{\sinh(\frac{i\beta}{2N}-u+\eta)}{\sinh(\frac{i\beta}{2N}-u)} \frac{\sinh(u+\frac{i\beta}{2N})}{\sinh(u+\frac{i\beta}{2N}-\eta)} + \mathcal{O}(\beta/N)^L \right)^N. \tag{4.40}$$

Taylor expanding the last factor to first order leads to

$$
\lim_{N \to \infty} \left( \frac{\sinh(\frac{i\beta}{2N} - u + \eta)}{\sinh(\frac{i\beta}{2N} - u)} \frac{\sinh(u + \frac{i\beta}{2N})}{\sinh(u + \frac{i\beta}{2N} - \eta)} + \ldots \right)^N =
$$
$$
\lim_{N \to \infty} \left( 1 + i(\coth(u) - \coth(u - \eta)) \frac{\beta}{N} + \mathcal{O}\left(\frac{\beta^2}{N^2}\right) \right)^N =
$$
$$
= \exp\left( i(\coth(u) - \coth(u - \eta))\beta \right).
$$

(4.41)

Substituting this back to the propagator, we get the final contour integral expression

$$
\langle k | e^{-iHt} | k - 1 \rangle = \oint_{\mathcal{C}} \frac{du}{2\pi i} \frac{\sinh(\eta)}{\sinh^2(-u + \eta)} \frac{1}{\frac{\sinh^L(u)}{\sinh^L(u - \eta)} - 1} \exp[i(\coth(u) - \coth(u - \eta))\beta].
$$

(4.42)

An interpretation of this formula together with the remaining cases will be given in 4.2.4.

### 4.2.2 One particle propagation by $\ell$ sites

We consider the propagator describing one particle moving $\ell$ sites, and the corresponding partition function in the Trotter decomposition:

$$
\langle k | e^{-iHt} | k - \ell \rangle \sim \text{Tr } D^{L-1-\ell} B D^{\ell-1} C.
$$

(4.43)

The computation of the product is very similar to the previous case, leading to the following result:

$$
D^{L-1-\ell} B D^{\ell-1} C = \sum_{i=1}^{n} c_i^2 b_i^{\ell-1} \sigma_i^- \sigma_i^+ \otimes_{j, j \neq i} \begin{pmatrix} b_j^L b_{ij}^{-1} & 0 \\ 0 & b_{ji}^{-1} \end{pmatrix}_{[j]} +
$$
$$
\sum_{i,j,i \neq j} c_i b_i^{\ell} \sigma_i^- \otimes c_j b_j^{L-1} \sigma_j^+ \otimes_{\substack{k \\ k \neq i,j}} \begin{pmatrix} b_k^L b_{jk}^{-1} & 0 \\ 0 & b_{ki}^{-1} \end{pmatrix}_{[k]}.
$$

(4.44)

For the trace we get:

$$
\text{Tr } D^{L-1-\ell} B D^{\ell-1} C = \sum_{i=1}^{n} c_i^2 b_i^{\ell-1} \prod_{j, j \neq i} \left( b_j^L b_{ij}^{-1} + b_{ji}^{-1} \right).
$$

(4.45)

This sum can be turned into a contour integral using the method of the previous subsection. We can apply the identity (4.36) with the same $g_j$ functions, but the following $f$:

$$
f(u) = \frac{b^{\ell-1}(-u) c^2(-u)}{\sinh(\eta)(b^L(-u) - 1)} = \frac{\sinh(\eta)}{\sinh^2(-u + \eta)} \frac{\sinh^{\ell-1}(u)}{\sinh^{\ell-1}(u - \eta)} \frac{1}{\frac{\sinh^L(u)}{\sinh^L(u - \eta)} - 1}.
$$

(4.46)

It can be seen that $f(u)$ is always regular at $u = 0$, and it is regular at $u = \eta$ if $\ell + 1 \leq L$. Therefore the contour integral representation based on (4.36) is valid only for $1 \leq \ell \leq L - 1$. This constraint is in agreement with our general picture about the lattice path integral: the propagator with $\ell = L$ corresponds to a particle staying at its position, due to periodicity. In this case a different trace needs to be evaluated which includes an $A$-operator, and this is treated in the next subsection.

Repeating the steps of the previous subsection we obtain for $\ell < L$

$$
\langle k | e^{-iHt} | k - \ell \rangle = \oint_{\mathcal{C}} \frac{du}{2\pi i} \frac{\sinh(\eta)}{\sinh^2(-u + \eta)} \frac{\sinh^{\ell-1}(u)}{\sinh^{\ell-1}(u - \eta)} \frac{\exp[i(\coth(u) - \coth(u - \eta))\beta]}{\frac{\sinh^L(u)}{\sinh^L(u - \eta)} - 1}.
$$

### 4.2.3   One particle not moving

Here we consider the case, when the particle stays in place. For this matrix element we have from (4.19):

$$\langle k|e^{-iHt}|k\rangle \sim \operatorname{Tr} D^{L-1}A. \tag{4.47}$$

The $A$ operator takes the following value at $u = 0$:

$$
A = \otimes_{i=1}^{n}\begin{pmatrix}1 & \\ & b_{i0}^{-1}\end{pmatrix}_{[i]} + \sum_{i=1}^{n} c_i^2 b_i^{-1}\sigma_i^-\sigma_i^+ \otimes_{j,j\neq i}\begin{pmatrix}b_j b_{ij}^{-1} & \\ & b_{ji}^{-1}\end{pmatrix}_{[j]} +
$$
$$
+ \sum_{\substack{i,j \\ i\neq j}} c_i b_{ji}^{-1}\sigma_i^- \otimes c_j \sigma_j^+ \otimes_{k,k\neq i,j}\begin{pmatrix}b_k b_{jk}^{-1} & \\ & b_{ki}^{-1}\end{pmatrix}_{[k]}. \tag{4.48}
$$

Out of the three terms of the $A$-operator, two contribute to the trace:

$$\operatorname{Tr} D^{L-1}A = \prod_{i=1}^{n}\left(b_i^{L-1}+b_{i0}^{-1}\right) + \sum_{i=1}^{n} c_i^2 b_i^{-1}\prod_{j,j\neq i} b_j^L b_{ij}^{-1} + b_{ji}^{-1}. \tag{4.49}$$

Note that the second term coincides with the expression of the previous subsection with $\ell = 0$. This suggests a close relation between the two cases, namely that the case of a particle staying at its place should be given directly by same formula, where $\ell = 0$. In the following we will see that this is indeed true, with the addition that the first term in (4.49) is also needed to produce the correct contour integral.

Consider again the product $f(u)\prod_j g_j(u)$, but in the present case assume that $f(u)$ has a simple pole at $\tilde{z}$ within the contour $\mathcal{C}$. Then we get

$$\oint_{\mathcal{C}}\frac{du}{2\pi i} f(u)\prod_j g_j(u) = \sum_{\xi_i} f(\xi_i)\left(\operatorname{Res}_{z=\xi_i} g_i(z)\right)\prod_{j,j\neq i} g_j(\xi_i) + \left(\operatorname{Res}_{z=\tilde{z}} f(u)\right)\prod_j g_j(\tilde{z}). \tag{4.50}$$

We choose

$$f(u) = \frac{c^2(-u)}{\sinh(\eta)}\frac{b^{-1}(-u)}{b^L(-u)-1} = \frac{\sinh(\eta)}{\sinh(-u+\eta)\sinh(-u)}\frac{\sinh^L(-u+\eta)}{\sinh^L(-u)-\sinh^L(-u+\eta)} \tag{4.51}$$
$$g_j(u) = b^L(-\xi_j)b^{-1}(u-\xi_j) + b^{-1}(\xi_j-u).$$

It can be seen that $f$ has a simple pole at $u = 0$ with residue one. Therefore, the identity (4.50) immediately gives the right hand side of (4.49).

Performing the Trotter limit as before we obtain the final contour integral

$$
\begin{aligned}
\langle k|e^{-iHt}|k\rangle &= \oint_{\mathcal{C}}\frac{du}{2\pi i} f(u)\exp\left(i(\coth(u)-\coth(u-\eta))\beta\right) \\
&= \oint_{\mathcal{C}}\frac{du}{2\pi i}\frac{\sinh(\eta)}{\sinh(u)\sinh(u-\eta)}\frac{1}{b^L(-u)-1}\exp\left(i(\coth(u)-\coth(u-\eta))\beta\right).
\end{aligned} \tag{4.52}
$$

As anticipated, this is formally identical to the result of the previous subsection with displacement $\ell = 0$.

### 4.2.4   Summary of the one particle formulas

A connection to the results of Section 3 can be given if we introduce a shift of $-\eta/2$ in the integration variable. Correspondingly, we also introduce the contour $\mathcal{C}_{\pm}$ that is a union of two

small circles around the points $\pm\eta/2$: $\mathcal{C}_\pm = \mathcal{C}_{-\eta/2} \cup \mathcal{C}_{\eta/2}$. Then the formulas of the previous subsections can be summarized as follows:

$$\langle k|e^{-iHt}|k-\ell\rangle = \oint_{\mathcal{C}_\pm} \frac{du}{2\pi i} q(u) \frac{P^\ell(u)}{1-P^L(u)} e^{-i\varepsilon(u)t}, \qquad \ell = 0 \ldots L-1. \tag{4.53}$$

Here $q(u)$, $P(u)$ and $\varepsilon(u)$ are defined in (3.34), (3.23) and (3.27), respectively. This coincides with the result obtain from the spectral representation.

## 4.3 The propagator: Two particle case

In this section we derive the two particle propagator. The computation is similar to the one particle case, nevertheless it poses some additional difficulties. In order to simplify the notations and later computations we introduce generalized operators that involve the action of $D$-operators from the left:

$$X^{(\ell)} \equiv D^\ell X, \qquad X = A, B, C. \tag{4.54}$$

Their explicit forms are

$$
\begin{aligned}
A^{(\ell)} \equiv D^\ell A &= \otimes_{i=1}^n \begin{pmatrix} b_i^\ell & \\ & b_{i0}^{-1} \end{pmatrix}_{[i]} + \sum_{i=1}^n c_i^2 b_i^{-1} \sigma_i^- \sigma_i^+ \otimes_{j,j\neq i} \begin{pmatrix} b_j^{\ell+1} b_{ij}^{-1} & \\ & b_{ji}^{-1} \end{pmatrix}_{[j]} \\
&\quad + \sum_{i,j,i\neq j} c_i c_j b_{ji}^{-1} b_j^\ell \sigma_i^- \otimes \sigma_j^+ \otimes_{\substack{k \\ k\neq i,j}} \begin{pmatrix} b_k^{\ell+1} b_{jk}^{-1} & \\ & b_{ki}^{-1} \end{pmatrix}_{[k]}
\end{aligned}
\tag{4.55}
$$

$$B^{(\ell)} := D^\ell B = \sum_{i=1}^n c_i \sigma_i^- \otimes_{j,j\neq i} \begin{pmatrix} b_j^{\ell+1} & 0 \\ 0 & b_{ji}^{-1} \end{pmatrix}_{[j]}$$

$$C^{(\ell)} := D^\ell C = \sum_{i=1}^n c_i b_i^\ell \sigma_i^+ \otimes_{j,j\neq i} \begin{pmatrix} b_j^{\ell+1} b_{ij}^{-1} & 0 \\ 0 & 1 \end{pmatrix}_{[j]}.$$

We will also use a simplified notation for the generalized operators, whenever we want to suppress the notation of the number of inserted $D$'s:

$$X^{(\ell)} \to \check{X}, \qquad X = A, B, C. \tag{4.56}$$

When considering products of generalized matrices in this notation, we assume, that the number of inserted $D$'s is generic and it can be put back to the formulas whenever needed.

In order to compute the propagator

$$\langle a, b|e^{-iHt}|c, d\rangle, \qquad \text{with} \quad 1 \le \begin{matrix} a < b \\ c < d \end{matrix} \le L, \tag{4.57}$$

we need to distinguish different cases depending on the relative position of the coordinates. According to the rule (4.13) and the cyclicity of the trace there are five different possibilities, that correspond to specific traces as follows.

- For $c < a < d < b$ one needs to compute $\text{Tr}\, B^{(\ell_1)} C^{(\ell_2)} B^{(\ell_3)} C^{(\ell_4)}$, where

$$\ell_1 = L + c - b - 1 \qquad \ell_2 = a - c - 1 \qquad \ell_3 = d - a - 1 \qquad \ell_4 = b - d - 1. \tag{4.58}$$

- For $c < a < b < d$ one needs to compute $\text{Tr}\, B^{(\ell_1)} C^{(\ell_2)} C^{(\ell_3)} B^{(\ell_4)}$, where

$$\ell_1 = L + c - d - 1 \qquad \ell_2 = a - c - 1 \qquad \ell_3 = b - a - 1 \qquad \ell_4 = d - b - 1. \tag{4.59}$$

- For $a = c < b < d$ one needs to compute $\mathrm{Tr}\, A^{(\ell_1)} C^{(\ell_2)} B^{(\ell_3)}$, where

$$\ell_1 = L + a - d - 1 = L + c - d - 1 \qquad \ell_2 = b - a - 1 = b - c - 1 \qquad \ell_3 = d - b - 1. \quad (4.60)$$

- For $a = c < d < b$ one needs to compute $\mathrm{Tr}\, A^{(\ell_1)} B^{(\ell_2)} C^{(\ell_3)}$, where

$$\ell_1 = L + a - b - 1 = L + c - d - 1 \qquad \ell_2 = d - a - 1 = d - c - 1 \qquad \ell_3 = b - d - 1. \quad (4.61)$$

- For $a = c < b = d$ one needs to compute $\mathrm{Tr}\, A^{(\ell_1)} A^{(\ell_2)}$, where

$$\ell_1 = L + a - b - 1 \qquad \ell_2 = b - a - 1. \quad (4.62)$$

For the following computations we introduce the notion of the *contracted traces*, which are similar to (but not identical with) the contractions used in the Wick theorem in free field theory. The motivation for the definition comes from the form of the $\check{X}$ operators: they are sums of products of operators such that in each product there are only one or two operators that act non-diagonally. After multiplying all $\check{X}$ and expanding the product, one can keep track of these non-diagonal operators by listing their indices, i.e. on which $\mathbb{C}^2$ subspaces of the QTM they act. The contracted trace is a particular sum of the traces of these products, where we sum over a specific index pattern while excluding coinciding indices. In the following we give examples for this idea.

For simplicity, we first consider the $B^{(\ell)}$ and $C^{(\ell)}$ operators. Only those terms of the product of $B^{(\ell)}$'s and $C^{(\ell)}$'s have non-vanishing trace, where an equal number of $\sigma^+$ and $\sigma^-$ share the same indices, and are ordered alternatingly. The simplest contracted traces are (using the notation (4.30))

$$\mathrm{Tr}\, \check{B}\check{C}_{(ii)} = \mathrm{Tr}\, \sum_i \mathcal{B}_i \mathcal{C}_i \otimes_{j,j \neq i} \beta_j^{(i)} \gamma_j^{(i)}$$
$$\mathrm{Tr}\, \check{C}\check{B}_{(ii)} = \mathrm{Tr}\, \sum_i \mathcal{C}_i \mathcal{B}_i \otimes_{j,j \neq i} \beta_j^{(i)} \gamma_j^{(i)}. \quad (4.63)$$

Here we take those $n$ terms from the expansion of the $\check{B}\check{C}$ (or $\check{C}\check{B}$) products where the non-diagonal part acts on site $i$, and afterwards we sum over these traces.

In the case of four operators we have more possibilities, for example

$$\mathrm{Tr}\, \check{B}\check{C}\check{B}\check{C}_{(iiii)} = \mathrm{Tr}\, \sum_i \mathcal{B}_i \mathcal{C}_i \mathcal{B}_i \mathcal{C}_i \otimes_{j,j \neq i} \beta_j^{(i)} \gamma_j^{(i)} \beta_j^{(i)} \gamma_j^{(i)}$$
$$\mathrm{Tr}\, \check{B}\check{C}\check{B}\check{C}_{(ijji)} = \mathrm{Tr}\, \sum_{\substack{i,j \\ i \neq j}} \mathcal{B}_i \gamma_i^{(j)} \beta_i^{(j)} \mathcal{C}_i \otimes \beta_j^{(i)} \mathcal{C}_j \mathcal{B}_j \gamma_j^{(i)} \otimes_{k,k \neq i,j} \beta_k^{(i)} \gamma_k^{(i)} \beta_k^{(j)} \gamma_k^{(j)}. \quad (4.64)$$

In the first case we take those $n$ terms where each non-diagonal part acts on site $i$, whereas in the second case the non-diagonal pieces act on sites $i$ and $j$ and we require $i \neq j$. This distinction is important: if we evaluate such a term for generic $i, j$ and set $i = j$ afterwards, we obtain a result that is different from the direct evaluation of the first case. Our strategy in the calculations is that we keep track of all possibilities explicitly and transform the sum of all terms into a contour integral.

A more complicated example of a contracted trace is

$$\mathrm{Tr}\, \check{B}\check{C}\check{B}\check{C}\check{B}\check{C}\check{C}\check{B}_{(iiiijjkk)} = \mathrm{Tr}\, \sum_{\substack{i,j,k \\ i \neq j \neq k}} \mathcal{B}_i \mathcal{C}_i \mathcal{B}_i \mathcal{C}_i \beta_i^{(j)} \gamma_i^{(j)} \gamma_i^{(k)} \beta_i^{(k)} \otimes$$
$$\otimes \beta_j^{(i)} \gamma_j^{(i)} \beta_j^{(i)} \gamma_j^{(i)} \mathcal{B}_j \mathcal{C}_j \gamma_j^{(k)} \beta_j^{(k)} \otimes \quad (4.65)$$
$$\otimes \beta_k^{(i)} \gamma_k^{(i)} \beta_k^{(i)} \gamma_k^{(i)} \beta_k^{(j)} \gamma_k^{(j)} \mathcal{C}_k \mathcal{B}_k \otimes$$
$$\otimes_{l,l \neq i,j,k} \beta_l^{(i)} \gamma_l^{(i)} \beta_l^{(i)} \gamma_l^{(i)} \beta_l^{(j)} \gamma_l^{(j)} \gamma_l^{(k)} \beta_l^{(k)}.$$

Incorporating the $\check{A}$ operators into this framework is straightforward based on (4.27): $\check{A}$ can be regarded as a sum of a diagonal term (first line in (4.27)) and a $\check{B}\check{C}$ product (second and third lines in (4.27)). When considering a contracted trace involving $\check{A}$ we have to specify, whether we refer to the diagonal or the $\check{B}\check{C}$ part. For terms that involve the diagonal piece we attach an empty set of indices to the $\check{A}$ operator and it will be denoted as $\check{A}_{()}$. For the non-diagonal terms there will be two indices attached to $\check{A}$ that specify the sites on which the two non-diagonal operators act. Examples for this will be shown in the subsections below and in Appendices C.2-C.3.

After these considerations we can give the definition for the contracted trace: The contracted trace is a sum of particular terms in the expansion of the trace of the product of $\check{X}$'s in the $F$-basis, and it is characterized by a specific index pattern. We sum over those terms in the expansion, where the off-diagonal factors of the various operators act on the subspaces designated by the index pattern. Furthermore, we exclude those terms from the sum where different indices would take coinciding values.

In this section we consider two configurations for the two particle propagator in detail (those corresponding to $\text{Tr } \check{B}\check{C}\check{B}\check{C}$ and $\text{Tr } \check{A}\check{C}\check{B}$); these two cases showcase all the computational nuances, which arise in the two particle case. Detailed calculations in the remaining three other cases are presented in C.1-C.3. A summary of the two-particle propagator is given in subsection 4.3.3.

### 4.3.1 The $\text{Tr } \check{B}\check{C}\check{B}\check{C}$ case

In the case of $\text{Tr } \check{B}\check{C}\check{B}\check{C}$ there are three non-vanishing contracted traces:

$$\text{Tr } \check{B}\check{C}\check{B}\check{C} = \text{Tr } \check{B}\check{C}\check{B}\check{C}_{(iiii)} + \text{Tr } \check{B}\check{C}\check{B}\check{C}_{(iijj)} + \text{Tr } \check{B}\check{C}\check{B}\check{C}_{(ijji)}. \tag{4.66}$$

These are computed as

$$\text{Tr } B^{(\ell_1)}C^{(\ell_2)}B^{(\ell_3)}C^{(\ell_4)}{}_{(iiii)} = \text{Tr } \left( \sum_{i=1}^n c_i^4 b_i^{\ell_2+\ell_4} \sigma_i^- \sigma_i^+ \sigma_i^- \sigma_i^+ \otimes_{j,j\neq i} \begin{pmatrix} b_j^{\ell_1+\ell_2+\ell_3+\ell_4+4} b_{ij}^{-2} & 0 \\ 0 & b_{ji}^{-2} \end{pmatrix}_{[j]} \right)$$

$$= \sum_i c_i^4 b_i^{\ell_2+\ell_4} \prod_{j,j\neq i} \left( b_j^L b_{ij}^{-2} + b_{ji}^{-2} \right)$$

$$\text{Tr } B^{(\ell_1)}C^{(\ell_2)}B^{(\ell_3)}C^{(\ell_4)}{}_{(iijj)} = \text{Tr } \sum_{i,j,i\neq j} c_i^2 b_i^{\ell_2} \sigma_i^- \sigma_i^+ \begin{pmatrix} b_i^{\ell_3+1} & \\ & b_{ij}^{-1} \end{pmatrix}_{[i]} \begin{pmatrix} b_i^{\ell_4+1} b_{ji}^{-1} & \\ & 1 \end{pmatrix}_{[i]} \otimes$$

$$\otimes c_j^2 b_j^{\ell_4} \begin{pmatrix} b_j^{\ell_1+1} & \\ & b_{ji}^{-1} \end{pmatrix}_{[j]} \begin{pmatrix} b_j^{\ell_2+1} b_{ij}^{-1} & \\ & 1 \end{pmatrix}_{[j]} \sigma_j^- \sigma_j^+ \otimes_{k,k\neq i,k\neq j}$$

$$\otimes_{k,k\neq i,k\neq j} \begin{pmatrix} b_k^{\ell_1+\ell_2+\ell_3+\ell_4+4} b_{ik}^{-1} b_{jk}^{-1} & \\ & b_{ki}^{-1} b_{kj}^{-1} \end{pmatrix}_{[k]}$$

$$= \sum_{i,j,i\neq j} c_i^2 c_j^2 b_i^{\ell_2} b_j^{\ell_4} b_{ij}^{-1} b_{ji}^{-1} \prod_{k,k\neq i,k\neq j} \left( b_k^L b_{ik}^{-1} b_{jk}^{-1} + b_{ki}^{-1} b_{kj}^{-1} \right)$$

$$\text{Tr } B^{(\ell_1)}C^{(\ell_2)}B^{(\ell_3)}C^{(\ell_4)}{}_{(ijji)} = \text{Tr } \sum_{i,j,i\neq j} c_i^2 b_i^{\ell_4} \sigma_i^- \begin{pmatrix} b_i^{\ell_2+1} b_{ji}^{-1} & \\ & 1 \end{pmatrix}_{[i]} \begin{pmatrix} b_i^{\ell_3+1} & \\ & b_{ij}^{-1} \end{pmatrix}_{[i]} \otimes$$

$$\otimes c_j b_j^{\ell_2} \begin{pmatrix} b_j^{\ell_1+1} & \\ & b_{ji}^{-1} \end{pmatrix}_{[j]} \sigma_j^+ \sigma_j^- \begin{pmatrix} b_j^{\ell_4+1} b_{ij}^{-1} & \\ & 1 \end{pmatrix}_{[j]} \otimes_{k,k\neq i,j}$$

$$\otimes_{k,k\neq i,j} \begin{pmatrix} b_k^{\ell_1+\ell_2+\ell_3+\ell_4+4} b_{jk}^{-1} b_{ik}^{-1} & \\ & b_{ki}^{-1} b_{kj^{-1}} \end{pmatrix}_{[k]}$$

$$= \sum_{i,j,i\neq j} c_i^2 c_j^2 b_i^{\ell_2+\ell_3+\ell_4+2} b_j^{\ell_4+\ell_1+\ell_2+2} b_{ji}^{-1} b_{ij}^{-1} \prod_{k,k\neq i,j} \left( b_k^L b_{ik}^{-1} b_{jk}^{-1} + b_{ki}^{-1} b_{kj}^{-1} \right). \tag{4.67}$$

Here we used $\ell_1 + \ell_2 + \ell_3 + \ell_4 + 4 = L$. For their sum we thus get

$$\text{Tr } B^{(\ell_1)}C^{(\ell_2)}B^{(\ell_3)}C^{(\ell_4)} = \sum_i c_i^4 b_i^{\ell_2+\ell_4} \prod_{j,j\neq i} \left( b_j^L b_{ij}^{-2} + b_{ji}^{-2} \right) +$$

$$+ \sum_{i,j,i\neq j} c_i^2 c_j^2 \, b_{ij}^{-1} b_{ji}^{-1} \left( b_i^{\ell_2} b_j^{\ell_4} + b_i^{\ell_2+\ell_3+\ell_4+2} b_j^{\ell_1+\ell_2+\ell_4+2} \right) \prod_{k,k\neq i,k\neq j} \left( b_k^L b_{ik}^{-1} b_{jk}^{-1} + b_{ki}^{-1} b_{kj}^{-1} \right). \tag{4.68}$$

The Trotter limit is taken after the identifications (4.21). As in the one particle case, this expression suffers from singularities: The $b_{ij}^{-1}$ type terms are singular in the $\beta_i \to \beta$, $\forall i$ limit. To overcome this, we use the same trick as for the one particle case. Consider the following double contour integral:

$$\oint_{C\times C} \oint \frac{du_1 du_2}{(2\pi i)^2} f(u_1, u_2) \prod_i \frac{h_i(u_1, u_2)}{g_i(u_1) g_i(u_2)}, \tag{4.69}$$

where $f$ and $h_i$ are meromorphic functions, which do not have poles inside $C$, and $1/g_i$ have simple poles at $\xi_1, \dots, \xi_n$ located inside the contour $C$. The integral can be evaluated by the successive application of the uni-variate residue formula:

$$\oint_{C\times C} \oint \frac{du_1 du_2}{(2\pi i)^2} f(u_1, u_2) \prod_i \frac{h_i(u_1, u_2)}{g_i(u_1) g_i(u_2)} =$$

$$= \sum_i f(\xi_i, \xi_i) h_i(\xi_i, \xi_i) \left( \text{Res}_{u_1=\xi_i} \frac{1}{g_i(u_1)} \right) \left( \text{Res}_{u_2=\xi_i} \frac{1}{g_i(u_2)} \right) \prod_{j,j\neq i} \frac{h_j(\xi_i, \xi_i)}{g_j(\xi_i) g_j(\xi_i)} +$$

$$+ \sum_{i,j,i\neq j} f(\xi_i, \xi_j) \frac{h_j(\xi_i, \xi_j)}{g_j(\xi_i)} \left( \text{Res}_{u_2=\xi_j} \frac{1}{g_j(u_2)} \right) \frac{h_i(\xi_i, \xi_j)}{g_i(\xi_j)} \left( \text{Res}_{u_1=\xi_i} \frac{1}{g_i(u_1)} \right) \times \tag{4.70}$$

$$\times \prod_{k\neq i,j} \frac{h_k(\xi_i, \xi_j)}{g_k(\xi_i) g_k(\xi_j)}.$$

The second summand of this expression is equal to $\left( \text{Tr } \check{B}\check{C}\check{B}\check{C}_{(iijj)} + \text{Tr } \check{B}\check{C}\check{B}\check{C}_{(ijji)} \right)$ if we specify the functions as

$$h_k(u_1, u_2) = b_k^L \sinh(u_1 - \xi_k + \eta) \sinh(u_2 - \xi_k + \eta) + \tag{4.71}$$
$$+ \sinh(\xi_k - u_1 + \eta) \sinh(\xi_k - u_2 + \eta)$$

$$g_k(u) = \sinh(u - \xi_k) \tag{4.72}$$

$$f(u_1, u_2) = f_{sym.}^{(\ell_2, \ell_4)}(u_1, u_2) + f_{sym.}^{(\ell_2+\ell_3+\ell_4+2, \ell_4+\ell_1+\ell_2+2)}(u_1, u_2), \tag{4.73}$$

with

$$f_{sym.}^{(x,y)}(u_1,u_2) = \frac{c^2(-u_1)c^2(-u_2)b^x(-u_1)b^y(-u_2)}{\sinh(\eta)\left(b^L(-u_2)S(u_2-u_1)-1\right)\sinh(\eta)\left(b^L(-u_1)S(u_1-u_2)-1\right)} \quad (4.74)$$

and $S(u)$ given by (3.30). It is easy to see that $f$ is indeed free of poles, given that $-1 < x < L-1$ and $-1 < y < L-1$.

If we substitute back these functions into the first summand of (4.70) then we obtain the first term of (4.66):

$$\sum_i f(\xi_i,\xi_i)h_i(\xi_i,\xi_i)\left(\mathrm{Res}_{u_1=\xi_i}\frac{1}{g_i(u_1)}\right)\left(\mathrm{Res}_{u_2=\xi_i}\frac{1}{g_i(u_2)}\right)\prod_{j,j\neq i}\frac{h_j(\xi_i,\xi_i)}{g_j(\xi_i)g_j^{(2)}(\xi_i)} =$$

$$= \sum_i c_i^4\left(b_i^{\ell_2}b_i^{\ell_4}+b_i^{\ell_2+\ell_3+\ell_4+2}b_i^{\ell_4+\ell_1+\ell_2+2}\right)\frac{1}{\sinh(\eta)\left(b_i^L\frac{\sinh(\eta)}{\sinh(-\eta)}-1\right)}\frac{1}{\sinh(\eta)\left(b_i^L\frac{\sinh(\eta)}{\sinh(-\eta)}-1\right)}\times$$

$$\times(b_i^L+1)\sinh^2(\eta)\prod_{j,j\neq i}\frac{b_j^L\sinh^2(\xi_j-\xi_i+\eta)+\sinh^2(\xi_i-\xi_j+\eta)}{\sinh^2(\xi_i-\xi_j)} =$$

$$= c_i^4 b_i^{\ell_2+\ell_4}\prod_{j,j\neq i}b_j^L b_{ij}^{-2}+b_{ji}^{-2} = \mathrm{Tr}\,\check{B}\check{C}\check{B}\check{C}_{(iiii)}.$$

Thus the double integral reproduces all three terms of (4.66) and we can write

$$\mathrm{Tr}\,B^{(\ell_1)}C^{(\ell_2)}B^{(\ell_3)}C^{(\ell_4)} = \oint_{\mathcal{C}\times\mathcal{C}}\oint\frac{du_1 du_2}{(2\pi i)^2}f(u_1,u_2)\prod_i\frac{h_i(u_1,u_2)}{g_i(u_1)g_i(u_2)}. \quad (4.75)$$

The propagator is obtained after we include the normalization factors:

$$\langle a,b|e^{-iHt}|c,d\rangle = \lim_{N\to\infty}\prod_{j=1}^{N}\left(\frac{\sinh\left(-\frac{i\beta_j}{2N}\right)}{\sinh\left(-\frac{i\beta_j}{2N}+\eta\right)}\right)^L\oint_{\mathcal{C}\times\mathcal{C}}\oint\frac{du_1 du_2}{(2\pi i)^2}f(u_1,u_2)\times$$

$$\times\prod_{j=1}^{2N}\frac{b^L(-\xi_j)\sinh(u_1-\xi_j+\eta)\sinh(u_2-\xi_j+\eta)+\sinh(\xi_j-u_1+\eta)\sinh(\xi_j-u_2+\eta)}{\sinh(u_1-\xi_j)\sinh(u_2-\xi_j)}, \quad (4.76)$$

valid for $c < a < d < b$.

The Trotter limit is taken analogously to the one particle case, leading to the result

$$\oint_{\mathcal{C}\times\mathcal{C}}\oint\frac{du_1 du_2}{(2\pi i)^2}f(u_1,u_2)\exp\left(i(\coth(u_1)+\coth(u_2)-\coth(u_1-\eta)-\coth(u_2-\eta))\beta\right). \quad (4.77)$$

Note, that taking the Trotter limit is completely independent of the function $f$.

A more transparent result is obtained if we express the $\ell_i, i = 1\ldots4$ with the original coordinates $a,b,c,d$ according to (4.58). Similar to the one-particle formula (4.53) we introduce a shift of $\eta/2$ and use the functions $q(u),P(u)$ and $S(u)$ (given by (3.34), (3.23) and (3.30)) to express the propagator as

$$\langle a,b|e^{-iHt}|c,d\rangle = \oint_{\mathcal{C}_\pm\times\mathcal{C}_\pm}\oint\frac{du_1 du_2}{(2\pi i)^2}\left(P^{a-c}(u_1)P^{b-d}(u_2)+P^{b-c}(u_1)P^{L+a-d}(u_2)\right)\times$$

$$\times q(u_1)q(u_2)\frac{\exp(-i(\varepsilon(u_1)+\varepsilon(u_2))t)}{(1-P^L(u_1)S(u_1-u_2))(1-P^L(u_2)S(u_2-u_1))} \quad (4.78)$$

(valid for $c < a < d < b$).

### 4.3.2 The Tr $\breve{A}\breve{C}\breve{B}$ case

Here we compute the case Tr $\breve{A}\breve{C}\breve{B}$ describing the propagator for $a = c < b < d$. The following contracted traces give non-vanishing contributions:

$$\text{Tr } \breve{A}\breve{C}\breve{B} = \text{Tr } \breve{A}\breve{C}\breve{B}_{(()ii)} + \text{Tr } \breve{A}\breve{C}\breve{B}_{(iijj)} + \text{Tr } \breve{A}\breve{C}\breve{B}_{(ijij)}. \tag{4.79}$$

As explained above, the empty indices () for the $\breve{A}$ operator denote that for term we take the diagonal part of $\breve{A}$, whereas in the other two cases we take the non-diagonal part and the two indices denote the sites on which the non-diagonal factors act.

These three contracted traces are computed as follows. We will use the relation $\ell_1 + \ell_2 + \ell_3 + 3 = L$ relevant to this case.

$$\begin{aligned}
\text{Tr } A^{(\ell_1)}C^{(\ell_2)}B^{(\ell_3)}{}_{(()ii)} &= \sum_{i=1}^{n} \begin{pmatrix} b_i^{\ell_1} & \\ & b_{i0}^{-1} \end{pmatrix}_{[i]} c_i^2 b_i^{\ell_2} \sigma_i^+ \sigma_i^- \otimes_{j,j\neq i} \begin{pmatrix} b_j^{\ell_1+\ell_2+\ell_3+2} b_{ij}^{-1} & \\ & b_{j0}^{-1} b_{ji}^{-1} \end{pmatrix}_{[j]} \\
&= \sum_{i=1}^{n} c_i^2 b_i^{\ell_1+\ell_2} \prod_{j,j\neq i} \left( b_j^L b_{0j}^{-1} b_{ij}^{-1} + b_{j0}^{-1} b_{ji}^{-1} \right)
\end{aligned} \tag{4.80}$$

$$\begin{aligned}
\text{Tr } A^{(\ell_1)}C^{(\ell_2)}B^{(\ell_3)}{}_{(iijj)} &= \sum_{\substack{i,j \\ i\neq j}} c_i^2 b_i^{-1} \sigma_i^- \sigma_i^+ \begin{pmatrix} b_i^{\ell_2+1} b_{ji}^{-1} & \\ & 1 \end{pmatrix}_{[i]} \begin{pmatrix} b_i^{\ell_3+1} & \\ & b_{ij}^{-1} \end{pmatrix}_{[i]} \otimes \\
&\quad \otimes \begin{pmatrix} b_j^{\ell_1+1} b_{ij}^{-1} & \\ & b_{ji}^{-1} \end{pmatrix}_{[j]} c_j^2 b_j^{\ell_2} \sigma_j^+ \sigma_j^- \otimes \\
&\quad \otimes_{k,k\neq i,j} \begin{pmatrix} b_k^{\ell_1+\ell_2+\ell_3+3} b_{ik}^{-1} b_{jk}^{-1} & \\ & b_{ki}^{-1} b_{kj}^{-1} \end{pmatrix}_{[k]} \\
&= \sum_{\substack{i,j \\ i\neq j}} c_i^2 c_j^2 b_i^{-1} b_j^{\ell_1+\ell_2+1} b_{ij}^{-1} b_{ji}^{-1} \prod_{k,k\neq i,j} \left( b_k^L b_{ik}^{-1} b_{jk}^{-1} + b_{ki}^{-1} b_{kj}^{-1} \right)
\end{aligned} \tag{4.81}$$

$$\begin{aligned}
\text{Tr } A^{(\ell_1)}C^{(\ell_2)}B^{(\ell_3)}{}_{(ijij)} &= \sum_{\substack{i,j \\ i\neq j}} c_i b_{ji}^{-1} \sigma_i^- c_i b_i^{\ell_2} \sigma_i^+ \begin{pmatrix} b_i^{\ell_3+1} & \\ & b_{ij}^{-1} \end{pmatrix}_{[i]} \otimes \\
&\quad \otimes c_j b_j^{\ell_1} \sigma_j^+ \begin{pmatrix} b_j^{\ell_2+1} b_{ij}^{-1} & \\ & 1 \end{pmatrix}_{[j]} c_j \sigma_j^- \otimes \\
&\quad \otimes_{k,k\neq i,j} \begin{pmatrix} b_k^{\ell_1+\ell_2+\ell_3+3} b_{jk}^{-1} b_{ik}^{-1} & \\ & b_{ki}^{-1} b_{kj}^{-1} \end{pmatrix}_{[k]} \\
&= \sum_{\substack{i,j \\ i\neq j}} c_i^2 c_j^2 b_i^{\ell_2} b_j^{\ell_1} b_{ji}^{-1} b_{ij}^{-1} \prod_{k,k\neq i,j} \left( b_k^L b_{ik}^{-1} b_{jk}^{-1} + b_{ki}^{-1} b_{kj}^{-1} \right).
\end{aligned} \tag{4.82}$$

In the first equation we used $b_j^{\ell_1+\ell_2+\ell_3+2} = b_j^L b_{0j}^{-1}$, where $b_{0j}^{-1} = b_j^{-1} = b^{-1}(-\xi_j)$. The full

partition function is thus

$$
\mathrm{Tr}\, A^{(\ell_1)} C^{(\ell_2)} B^{(\ell_3)} = \sum_{i=1}^{n} c_i^2 b_i^{\ell_1+\ell_2} \prod_{j,j\neq i} \left( b_j^L b_{0j}^{-1} b_{ij}^{-1} + b_{j0}^{-1} b_{ji}^{-1} \right) +
$$
$$
+ \sum_{\substack{i,j \\ i\neq j}} c_i^2 c_j^2 \left( b_i^{-1} b_j^{\ell_1+\ell_2+1} b_{ij}^{-2} + b_i^{\ell_2} b_j^{\ell_1} b_{ij}^{-1} b_{ji}^{-1} \right) \prod_{k,k\neq i,j} \left( b_k^L b_{ik}^{-1} b_{jk}^{-1} + b_{ki}^{-1} b_{kj}^{-1} \right).
$$

$$(4.83)$$

This expression is similar to (4.68), but we notice a few differences:

- There is no $\mathrm{Tr}\, \check{A}\check{C}\check{B}_{(iiii)}$ term, because this contracted trace is automatically zero. By its structure this term would correspond to $\mathrm{Tr}\, \check{B}\check{C}\check{B}\check{C}_{(iiii)}$ from the previous case.

- On the other hand, there is an extra term $\mathrm{Tr}\, \check{A}\check{C}\check{B}_{(()ii)}$ which does not have corresponding term in the $\mathrm{Tr}\, \check{B}\check{C}\check{B}\check{C}$ case. Based on the one particle case (Section 4.2.3) we can anticipate the role of this term on the contour integral side: The $f$ function will not be free of poles within $\mathcal{C}$, and considering the pole of $f$ will lead to this term.

- In the previous case only symmetric products of $b_{ij}^{-1} b_{ji}^{-1}$ occurred, for both $\mathrm{Tr}\, \check{B}\check{C}\check{B}\check{C}_{(iijj)}$ and $\mathrm{Tr}\, \check{B}\check{C}\check{B}\check{C}_{(ijji)}$. However, in this case there is also a non-symmetric $b_{ij}^{-2}$ present.

All these differences find an explanation as we transform the sums into a common contour integral.

Consider again the integral $\oint \oint \frac{du_1 du_2}{(2\pi i)^2} f(u_1,u_2) \prod_i \frac{h_i(u_1,u_2)}{g_i(u_1)g_i(u_2)}$ with the assumption that $f(u_1,u_2)$ has a pole at $u_1 = \tilde{z}$:

$$
\oint_{\mathcal{C}\times\mathcal{C}} \oint \frac{du_1 du_2}{(2\pi i)^2} f(u_1,u_2) \prod_i \frac{h_i(u_1,u_2)}{g_i(u_1)g_i(u_2)} =
$$
$$
= \sum_i f(\xi_i,\xi_i) h_i(\xi_i,\xi_i) \left( \mathrm{Res}_{u_1=\xi_i} \frac{1}{g_i(u_1)} \right) \left( \mathrm{Res}_{u_2=\xi_i} \frac{1}{g_i(u_2)} \right) \prod_{j,j\neq i} \frac{h_j(\xi_i,\xi_i)}{g_j(\xi_i)g_j(\xi_i)} +
$$
$$
+ \sum_{i,j,i\neq j} f(\xi_i,\xi_j) \frac{h_j(\xi_i,\xi_j)}{g_j(\xi_i)} \left( \mathrm{Res}_{u_2=\xi_j} \frac{1}{g_j(u_2)} \right) \frac{h_i(\xi_i,\xi_j)}{g_i(\xi_j)} \left( \mathrm{Res}_{u_1=\xi_i} \frac{1}{g_i(u_1)} \right) \times
$$
$$
\times \prod_{k\neq i,j} \frac{h_k(\xi_i,\xi_j)}{g_k(\xi_i)g_k(\xi_j)} + \sum_{i=1}^{n} \left( \mathrm{Res}_{u_1=\tilde{z}} f(\tilde{z},\xi_i) \right) \frac{h_i(\tilde{z},\xi_i)}{g_i(\tilde{z})} \left( \mathrm{Res}_{u_2=\xi_i} \frac{1}{g_i(u_2)} \right) \prod_{j,j\neq i} \frac{h_j(\tilde{z},\xi_i)}{g_j(\tilde{z})g_j(\xi_i)}.
$$

$$(4.84)$$

The second summand can be identified with $\mathrm{Tr}\, \check{A}\check{C}\check{B}_{(iijj)} + \mathrm{Tr}\, \check{A}\check{C}\check{B}_{(ijij)}$ if we specify

$$
h_k(u_1,u_2) = b_k^L \sinh(u_1 - \xi_k + \eta)\sinh(u_2 - \xi_k + \eta) + 
$$
$$
+ \sinh(\xi_k - u_1 + \eta)\sinh(\xi_k - u_2 + \eta) \tag{4.85}
$$

$$
f(u_1,u_2) = f_{non-sym.}^{(-1,\ell_1+\ell_2+1)}(u_1,u_2) + f_{sym.}^{(\ell_2,\ell_1)}(u_1,u_2) \tag{4.86}
$$

$$
f_{non-sym.}^{(x,y)}(u_1,u_2) = \frac{c^2(-u_1)c^2(-u_2)S(u_2-u_1)b^x(-u_1)b^y(-u_2)}{\sinh(\eta)(b^L(-u_2)S(u_2-u_1)-1)\sinh(\eta)(b^L(-u_1)S(u_1-u_2)-1)} \tag{4.87}
$$

and $g_k(u)$, $f_{sym.}^{(x,y)}(u_1,u_2)$ and $S(u)$ given by (4.72), (4.74) and (3.30) respectively.

The difference between $f_{non-sym.}$ and $f_{sym}$ is merely the extra factor of $S(u_2-u_1)$. This factor has a pole for $u_2 - u_1 + \eta = 0$, and such points are included in the double contour

integral. However, this pole is canceled by the same factor appearing also in the denominator, therefore it does not give any new terms.

After this identification the first term of (4.84) is evaluated as

$$
\begin{aligned}
\sum_{i=1}^{n} f(\xi_i, \xi_i) h_i(\xi_i, \xi_i) &\prod_{j, j \neq i} \frac{h_j(\xi_i, \xi_i)}{g_j(\xi_i) g_j(\xi_i)} = \\
&= \sum_{i=1^n} c_i^4 \frac{1}{\sinh^2(\eta) \left(b_i^L(-1) - 1\right)^2} \left((-1) b_i^{\ell_1 + \ell_2} + b_i^{\ell_1 + \ell_2}\right) \left(b_i^L \sinh^2(\eta) + \sinh^2(\eta)\right) \times \\
&\times \prod_{j, j \neq i} \frac{b_j^L \sinh^2(\xi_i - \xi_j + \eta) + \sinh^2(\xi_j - \xi_i + \eta)}{\sinh^2(\xi_i - \xi_j)} = 0.
\end{aligned}
\tag{4.88}
$$

Here we used the identity $S(0) = -1$. The main reason for the vanishing of this expression is that for this particular contracted trace the functions $f_{non-sym.}$ and $f_{sym.}$ cancel each other.

Finally we treat the third term in (4.84). The function $f(u_1, u_2)$ is singular at $u_1 = 0$ with the residue given by

$$
\begin{aligned}
\mathrm{Res}_{u_1=0} f(u_1, \xi_j) &= \mathrm{Res}_{u_1=0} f_{non-sym.}^{(-1, \ell_1 + \ell_2 + 1)}(u_1, \xi_i) = \\
&\frac{c_i^2}{\sinh(\eta)} \frac{\sinh(-\xi_i + \eta)}{\sinh(-\xi_i - \eta)} \frac{\sinh^{\ell_1 + \ell_2 + 1}(-\xi_i)}{\sinh^{\ell_1 + \ell_2 + 1}(-\xi_i + \eta)} \frac{1}{b_i^L \frac{\sinh(-\xi_i + \eta)}{\sinh(-\xi_i - \eta)} - 1}.
\end{aligned}
\tag{4.89}
$$

Collecting all factors we can see that the third term in (4.84) reproduces the term $\mathrm{Tr}\,\check{A}\check{C}\check{B}_{(\,)ii}$ in (4.79). Thus we get the full equality:

$$
\mathrm{Tr}\,\check{A}\check{C}\check{B} = \oint_{\mathcal{C} \times \mathcal{C}} \oint \frac{du_1 du_2}{(2\pi\mathrm{i})^2} f(u_1, u_2) \prod_i \frac{h_i(u_1, u_2)}{g_i(u_1) g_i(u_2)},
\tag{4.90}
$$

with functions defined in (4.85), (4.72) and (4.87).

Taking the Trotter limit follows the same steps as previously. We introduce the shift of $-\eta/2$ once again and obtain the final result

$$
\begin{aligned}
\langle a, b | e^{-\mathrm{i}Ht} | c, d \rangle = \oint_{\mathcal{C}_\pm \times \mathcal{C}_\pm} \oint \frac{du_1 du_2}{(2\pi\mathrm{i})^2} &\left(S(u_2 - u_1) P^{L+b-d}(u_2) + P^{b-c}(u_1) P^{L+a-d}(u_2)\right) \times \\
&\times q(u_1) q(u_2) \frac{\exp\left(-\mathrm{i}(\varepsilon(u_1) + \varepsilon(u_2))t\right)}{(1 - P^L(u_1) S(u_1 - u_2))(1 - P^L(u_2) S(u_2 - u_1))}
\end{aligned}
\tag{4.91}
$$

(valid for $c = a < b < d$).

### 4.3.3 Summary of two particle case

In the previous subsections we have computed two out of the five cases for the two-particle propagator. The other three cases can be treated similarly, and the detailed computations are presented in appendices C.1 to C.3. The common properties of these calculations are the following: The partition function at finite $N$ is transformed into a double contour integral. The $h_k$ and $g_k$ functions are the same in all cases, and $f$ depends on the specific configuration. The various contracted traces correspond to various terms in the sum over residues.

The derivations lead to the following general form:

$$
\langle a, b | e^{-\mathrm{i}Ht} | c, d \rangle = \oint_{\mathcal{C}_\pm \times \mathcal{C}_\pm} \oint \frac{du_1 du_2}{(2\pi\mathrm{i})^2} q(u_1) q(u_2) \frac{\Psi_{\{a,b\},\{c,d\}}(u_1, u_2) \exp\left(-\mathrm{i}(\varepsilon(u_1) + \varepsilon(u_2))t\right)}{(1 - P^L(u_1) S(u_1 - u_2))(1 - P^L(u_2) S(u_2 - u_1))}.
$$

$$
\tag{4.92}
$$

Here $\Psi_{\{a,b\},\{c,d\}}(u_1, u_2)$ is an amplitude, with the explicit form in the five different cases being

$$
\Psi_{\{a,b\},\{c,d\}}(u_1, u_2) =
$$
$$
= \begin{cases}
P^{a-c}(u_1)P^{b-d}(u_2) + P^{b-c}(u_1)P^{L+a-d}(u_2) & \text{if } c < a < d < b \ (\text{Tr } \check{B}\check{C}\check{B}\check{C}) \\
S(u_2 - u_1)P^{a-c}(u_1)P^{L+b-d}(u_2) + P^{b-c}(u_1)P^{L+a-d}(u_2) & \text{if } c < a < b < d \ (\text{Tr } \check{B}\check{C}\check{C}\check{B}) \\
S(u_2 - u_1)P^{L+b-d}(u_2) + P^{b-c}(u_1)P^{L+a-d}(u_2) & \text{if } a = c < b < d \ (\text{Tr } \check{A}\check{C}\check{B}) \\
P^{b-d}(u_2) + P^{b-c}(u_1)P^{L+a-d}(u_2) & \text{if } a = c < d < b \ (\text{Tr } \check{A}\check{B}\check{C}) \\
1 + P^{b-c}(u_1)P^{L+a-d}(u_2) & \text{if } a = c < b = d \ (\text{Tr } \check{A}\check{A}).
\end{cases}
$$
$$(4.93)$$

These five formulas emerged from the concrete computations, and they take a different form than the results of Section 3. First we give an interpretation of these formulas, and afterwards we explain how they can be compared to the earlier multiple integrals.

We can see that the amplitude $\Psi_{\{a,b\},\{c,d\}}(u_1, u_2)$ above is reminiscent of the Bethe Ansatz wave function, but it is not identical to it. It depends on both the initial and the final coordinates. It is given by a sum over permutations, where we permute the final positions of two particles, which are started from positions $c$ and $d$ and are indexed with their rapidity parameters $u_1$ and $u_2$, respectively. For each permutation there is an assigned phase, which consists of the one-particle propagation phases and it can also include a scattering phase. We observe that the phases $P^\ell(u)$ with some $\ell$ associated to the one-particle propagation are such that each particle always travels to the right, and if its final position is to the left of the initial one, then it is required that the particle travels around the volume. It is important that this rule is not imposed by any Ansatz, rather it emerges naturally from the computation. Also, this rule of "moving to the right" is not physical and it is only used to construct $\Psi_{\{a,b\},\{c,d\}}(u_1, u_2)$. Regarding the scattering phases we can observe that a factor of $S(u_2 - u_1)$ is inserted if during that particular displacement process if there is a crossing of the world lines of the two particles. A pictorial interpretation of these displacement processes and the construction of the amplitude is given in Fig. 5.

We remind that the five cases listed above do not exhaust all possible initial and final positions, and all other possibilities follow from periodicity, which is implied by the periodicity of the traces (4.13).

## 4.4 Connection to the results from the spectral series

The formulas (4.92)-(4.93) display a manifest periodicity in the space coordinates, and this follows simply from the properties of the traces (4.19). Nevertheless they can be easily connected the results of Section 3.

To do this let us consider again the spectral expansion (3.41), where the summation runs over all the Bethe states of the $\kappa$-deformed model. The amplitude $W^{(\kappa)}_{\{b\},\{a\}}$ under the multiple integral can be transformed in many ways by substituting a subset of the Bethe equations. This way we can exploit the fact that the eigenfunctions are periodic, thus constructing an amplitude which is manifestly periodic in both sets of coordinates. This procedure does not introduce new singular points, and therefore the contour manipulations can be carried out in exactly the same way, by expressing the sum over the Bethe states as a single contour integral around the simplified contour $\mathcal{C}_\pm$. It can be seen that all 5 cases detailed in (4.93) can be obtained in this way, by using also the permutation symmetry of the integrals (3.44).

## 4.5 The multi-particle case

The multi-particle situation can be treated similar to the two-particle case detailed above, but the evaluation of the traces and the derivation of the contour-integrals bears considerable

The case of $c < a < d < b$:

$$\Psi_{\{a,b\},\{c,d\}}(u_1, u_2) = \qquad P^{a-c}(u_1)P^{b-d}(u_2) + \qquad P^{b-c}(u_1)P^{L+a-d}(u_2)$$

The case of $c < a < b < d$:

$$\Psi_{\{a,b\},\{c,d\}}(u_1, u_2) = \qquad S(u_2 - u_1)P^{a-c}(u_1)P^{L+b-d}(u_2) + \quad P^{b-c}(u_1)P^{L+a-d}(u_2)$$

The case of $a = c < b < d$:

$$\Psi_{\{a,b\},\{c,d\}}(u_1, u_2) = \qquad S(u_2 - u_1)P^{L+b-d}(u_2) + \qquad P^{b-c}(u_1)P^{L+a-d}(u_2)$$

Figure 5: A pictorial interpretation for the amplitude $\Psi_{\{a,b\},\{c,d\}}(u_1, u_2)$ in three different cases. The particles 1 and 2 start from the initial positions $c$ and $d$, respectively, and the amplitude is given by two terms corresponding to the two possibilities of occupying the final positions $a$ and $b$. The propagation phases are obtained by counting the number of sites that the particles move to the right. A scattering phase $S(u_2 - u_1)$ is added when there is a crossing of the world-lines.

technical difficulties. It is possible to derive a formula similar to (4.92) with an amplitude which is manifestly periodic, which could then be transformed into the form (3.47).

We have not found a simple combinatorial proof within our F-basis computations, and in view of the existing relatively simple result (3.47) we refrain from including the long combinatorial proof here.

# 5 The Loschmidt amplitude for the domain wall quench

As an application of the previous results here we consider the physical situation when the initial state is the so-called domain wall state, consisting of $m$ down spins embedded in a volume of length $L$:

$$\left|DW_{L,m}\right\rangle \equiv \left|\underbrace{\downarrow \ldots \downarrow}_{m \text{ times}} \underbrace{\uparrow \ldots \uparrow}_{L-m \text{ times}}\right\rangle. \tag{5.1}$$

The arising dynamics has already been studied using Algebraic Bethe Ansatz [75], DMRG [76], the Generalized Hydrodynamics [77], and a special method building on the integrability of the model [78].

The simplest object to compute is the so-called Loschmidt amplitude or return amplitude:

$$\mathcal{L}(t) = \left\langle DW_{L,m}\big|e^{-iHt}\big|DW_{L,m}\right\rangle. \tag{5.2}$$

This object has been computed directly in the thermodynamic limit (sending both $m$ and $L$ to infinity) in [78]. Here we compute an exact finite volume representation for the return amplitude.

The object (5.2) can be expanded directly into a spectral series as

$$\mathcal{L}(t) = \sum_j \frac{|\langle DW_{L,m}|\Psi_j\rangle|^2}{\langle \Psi_j|\Psi_j\rangle} e^{-iE_j t}. \tag{5.3}$$

It is known that the scalar product of a Bethe state and the domain wall state is given by the so-called Izergin-Korepin (IK) determinant [75,79,80]. This representation would thus lead to a multiple integral over a squared IK determinant. On the other hand, our representation (3.47) for the propagator involves a product of a Bethe wave function and a free wave function, thus $\mathcal{L}(t)$ can be expressed as some integral involving only a single IK determinant.

Let us consider the scalar product

$$\langle 0|\prod_{j=1}^m C(\lambda_j - \eta/2)|DW_{L,m}\rangle, \tag{5.4}$$

where the $C$-operators are defined in the normalization given by (3.2)-(3.4). It can be seen on the explicit form of the wave functions (3.21) that this scalar product is independent of $L$, because the excitations only occupy the positions $1, 2, \ldots, m$. Therefore, this scalar product is given by the corresponding expression at $L = m$:

$$\tilde{Z}_m(\{\lambda\}) \equiv \langle 0|\prod_{j=1}^m C(\lambda_j - \eta/2)|DW_{m,m}\rangle. \tag{5.5}$$

This object can be expressed using the Izergin-Korepin determinant [80]. However, the IK determinant is first derived for an inhomogeneous spin chain with inhomogeneity paramaters $\mu_k$, $k = 1, \ldots, m$ [75,80]. In that case the corresponding scalar product is

$$\tilde{Z}_m(\{\lambda\}, \{\mu\}) = \frac{\prod_{j,k} \sinh(\lambda_j - \mu_k - \eta/2)}{\prod_{j>k} \sinh(\lambda_j - \lambda_k)\sinh(\mu_k - \mu_j)} \det T, \tag{5.6}$$

where

$$T_{jk} = t(\lambda_j - \mu_k), \qquad t(u) = \frac{\cosh(u - \eta/2)}{\sinh(u - \eta/2)} - \frac{\cosh(u + \eta/2)}{\sinh(u + \eta/2)}. \tag{5.7}$$

Performing the homogeneous limit $\mu_j \to 0$ we get

$$\tilde{Z}_m(\{\lambda\}) = (-1)^{m(m-1)/2} \frac{\prod_j \sinh^m(\lambda_j - \eta/2)}{\prod_{j>k} \sinh(\lambda_j - \lambda_k)} \det \bar{T}, \tag{5.8}$$

with

$$\bar{T}_{jk} = \coth^k(\lambda_j - \eta/2) - \coth^k(\lambda_j + \eta/2). \tag{5.9}$$

As explained above, formula (5.8) describes the overlap (5.4) for arbitrary $L$ and therefore it can be substituted into our multiple integral formula (3.47).

Collecting all additional factors coming from the normalization of the wave functions, and adding also the free part of the propagator amplitude we get the final formula for the Loschmidt amplitude

$$\langle DW_m|e^{-iHt}|DW_m\rangle = \prod_{j=1}^m \left(\oint_{\mathcal{C}} \frac{du_j}{2\pi i} q(u_j)\right) \times \prod_j P^j(u_j) \times$$

$$\times \frac{\prod_j \sinh^{m+1}(u_j - \eta/2)}{\sinh^m(\eta) \prod_{j<k} \sinh(u_j - u_k - \eta)} \det \bar{T} \frac{e^{-i(\sum_{k=1}^m \varepsilon(u_k))t}}{\prod_{k=1}^m \left(1 - e^{Q_j(\{u\})}\right)}. \tag{5.10}$$

We have implemented this formula for low particle number $N = 2$ and we have checked that it indeed reproduces the Loschmidt echo obtained from exact diagonalization for various values of $L$. It would be interesting to compute the asymptotic value of $\mathcal{L}(t)$ in the limit $N, L \to \infty$, and possibly to derive the sub-leading corrections too. This would lead to an independent confirmation of the results of [78].

# 6  Conclusions and Discussion

We have obtained a multiple integral representation for the propagator of the finite volume XXZ chain. Our main result is formula (3.47), which is a compact expression that uses well-defined functions and contours for every $L$ and $m$, such that the volume enters simply as a parameter. This representation is very similar to the multiple integral formulas derived earlier for the equilibrium correlation functions in [6, 7].

We applied two methods for the computation. The direct spectral sum quickly leads to a compact representation, whereas our second method through the QTM and its F-basis is much more complicated. Nevertheless it is interesting that this method leads to an amplitude for the multiple integrals, which is manifestly periodic for an arbitrary set of rapidities.

Having found a compact representation for the propagator, it is important to discuss the practical applicability of the result. The $m$-particle propagator is given by an $m$-fold integral, and numerical implementations become very quickly unfeasible as we increase $m$. At present it seems that for most numerical purposes the known approximate methods (for example t-DMRG or ABACUS) or even exact diagonalization would perform better than the numerically exact evaluation of the multiple integral. Advantages of our representation could show up in the study of the long time limit or finite size effects at large $L$ (see below).

The propagator is an intermediate object that can be used to compute the time dependent local observables through the sum

$$\langle \Psi_0 | \mathcal{O}(t) | \Psi_0 \rangle =$$
$$= \sum_{\{a\},\{b\},\{c\},\{d\}} \langle \Psi_0 | \{d\} \rangle \langle \{d\} | e^{iHt} | \{c\} \rangle \langle \{c\} | \mathcal{O} | \{b\} \rangle \langle \{b\} | e^{-iHt} | \{a\} \rangle \langle \{a\} | \Psi_0 \rangle.$$

This is an alternative to the usual spectral representation: here the (double) sum over the Bethe states is included in the exact propagator, and the remaining task is to perform the real space summations. In many practical applications (in concrete quench protocols) the initial states take simple forms in the real space representations, thus the only challenging task is to perform the inner sums that connect the two propagators to the local operator. Here one needs to have good control over the overlaps with the ininitial states.

In the present work we treated the overlaps with the domain wall state and computed the Loschmidt amplitude. An extension to other initial states is left to further research.

Once the overlaps have been added into these computations, it would be desirable to compute the asymptotic behaviour of the multiple integrals. This could lead to a direct verification of the predictions of the Generalized Hydrodynamics (GHD) [24–28]. However, this task is very involved. Regarding the equilibrium correlations the asymptotics in the static case has been treated successfully starting from the formulas of [6] (see [9, 12, 81]), but the dynamical case remained open. At present it is not clear, whether such computation is possible for the non-equilibrium problems treated in our work.

# Acknowledgments

The authors would like to thank Frank Göhmann, Gábor Takács, Véronique Terras, and Michael Wheeler for valuable discussion and comments. Also, we would like to thank an anonymous referee for bringing the works [6,7] to our attention, which led to an improvement of our paper, in particular regarding the evaluation of the spectral sum for the propagator.

This research was supported by the BME-Nanotechnology FIKP grant of EMMI (BME FIKP-NAT), by the National Research Development and Innovation Office (NKFIH) (K-2016 grant no. 119204, the OTKA grant no. SNN118028, and the KH-17 grant no. 125567), and by the "Premium" Postdoctoral Program of the Hungarian Academy of Sciences. GZF would like to thank the hospitality of the mathematical research institute MATRIX in Australia, where part of the research was carried out.

# A  Multidimensional residues

Let us consider $\mathbb{C}^N$ and $N$ meromorphic functions $g_k : \mathbb{C}^N \to \mathbb{C}$ such that each of them has a zero at the point $\mathbf{z} = (x_1, x_2, \ldots, x_N)$:

$$g_k(x_1, \ldots, x_N) = 0, \ldots k = 1, \ldots, N. \tag{A.1}$$

In this case there is a multi-dimensional residue theorem for the contour integrals around this singular point, but the precise form of the statement differs from the one dimensional Cauchy theorem in certain respects.

Let us construct the meromorphic $N$-form

$$d\mathbf{z} \equiv dz_1 \wedge dz_2 \wedge \cdots \wedge dz_N, \tag{A.2}$$

where $\wedge$ stands for the outer product.

We construct an $N$-dimensional real surface $\Gamma_{\mathbf{g}}$ in $\mathbb{C}^N$ for some small parameters $(\varepsilon_1, \ldots, \varepsilon_N)$, $\varepsilon_j \in \mathbb{R}^+$:

$$\Gamma_{\mathbf{g}} = \{\mathbf{z} \in \mathbb{C}^N, |g_j(\mathbf{z})| = \varepsilon_j\}. \tag{A.3}$$

The general multi-dimensional residue statement is the following [82]. For sufficiently small parameters and arbitrary holomorphic function $f : \mathbb{C}^N \to \mathbb{C}$ we have

$$\int_{\Gamma_{\mathbf{g}}} \frac{d\mathbf{z}}{(2\pi i)^N} \frac{f}{g_1 g_2 \ldots g_N} = \frac{f(\{x\})}{\det J_{jk}}, \tag{A.4}$$

where the matrix $J$ is defined as

$$J_{jk} = \frac{\partial g_j}{\partial z_k}\bigg|_{\{z\}=\{x\}}. \tag{A.5}$$

Notice that both the left and the right hand sides are anti-symmetric with respect to an exchange of variables. On the l.h.s. this follows from the anti-symmetry of the integration measure, whereas on the r.h.s. this is a property of the determinant.

It is important that here the contour $\Gamma_{\mathbf{g}}$ depends on the $g_k$ functions, and it is not predefined by the coordinates: it is necessary that in the multiple integral each $g_k$ "winds around" exactly one time.

The multiple integrals with pre-defined contours might lead to unexpected results. Consider for example the double integral

$$I(a, b, c, d) \equiv \oint_C \frac{dx}{2\pi i} \oint_C \frac{dy}{2\pi i} \frac{1}{(ax + by)(cx + dy)}, \qquad |a| \neq |b|, \quad |c| \neq |d|, \tag{A.6}$$

where for simplicity we choose both contours $C$ to be a unit circle around zero. A direct evaluation of this double integral is possible after partial fraction decomposition, which leads to the result

$$I = \frac{1}{ad - bc} \left[ \delta_{|c|<|d|} - \delta_{|a|<|b|} \right]. \tag{A.7}$$

We can see that the integral depends crucially on the ratios of the elements of the matrix $J$: it can produce $\pm \frac{1}{\det J}$, but also zero. The reason for this is that depending on the parameters the two functions $g_1 = ax + by$, $g_2 = cx + dy$ might not have winding number 1 around 0. On the other hand, the expected result (A.4) is reproduced with the contours with equal radiuses if the matrix $J$ is dominated by the diagonal elements.

It is also useful to consider the integral for contours with different radiuses $r_{1,2}$. We obtain the result

$$I = \frac{1}{ad - bc} \left[ \delta_{|cr_1|<|dr_2|} - \delta_{|ar_1|<|br_2|} \right]. \tag{A.8}$$

### A.1 The spectral sum in the $\kappa \to 0$ limit

In Section 3 we evaluated the spectral sum for some $\kappa$ twist parameters close enough to 0. The reason for this was that in this limit the completeness of the Bethe Ansatz can be rigorously proven in a relatively simple way; this was performed in [7]. Here we do not repeat the computations of [7], we merely summarize the essential points and explain how the contour integral manipulations fit together with the general formula (A.4).

It can be seen from (3.24) that with a fixed $L$ and for a small enough $\kappa$ the Bethe roots will cluster around the special point $-\eta/2$. It was shown in [7] that in this limit we have as many admissible solutions as required for the completeness of the Bethe Ansatz, which also uses a statement about the linear independence of the Bethe vectors with different sets of roots.

The summation over Bethe states can be expressed as integrals around the Bethe roots as in formula (3.41). It is important for this computation to have a good control over the elements of the Gaudin matrix. It follows from the explicit formula (3.33) that if all Bethe roots are close to $-\eta/2$ then the matrix is indeed dominated by the diagonal elements, due to the divergent $q(u)$ functions. Thus in these cases the general multidimensional integral formula is indeed equivalent to the integrals of the form (3.41).

### A.2 Singular states for $N = 2$

It is known that the untwisted Hamiltonian (2.1) can have physical eigenstates that are described by so-called singular Bethe states. They are described by non-admissible solutions of the Bethe equations (3.24), and the set of rapidities includes the singular values $\pm \eta/2$. Such solutions exist in even volumes $L \geq 4$ and have been studied extensively [62, 63, 83–86].

A special case of the singular solutions is a two-particle state which for the homogeneous chain consists of the pair

$$\{\lambda_1, \lambda_2\} = \{\eta/2, -\eta/2\}. \tag{A.9}$$

These rapidities form a perfect two-string, and correspondingly the state can be interpreted as an infinitely bound state of two spin waves. The wave function is proportional to [84]

$$\sum_j (-1)^j \sigma_j^- \sigma_{j+1}^- |0\rangle. \tag{A.10}$$

The paper [63] discussed the deformation of this state when a twist parameter is introduced. The rapidities (A.9) are always non-admissible solutions to the Bethe equations (3.24), for every finite twist $\kappa$, and they do not correspond to physical eigenstates of the twisted Hamiltonian (3.15). On the other hand, there is a $\kappa$-dependent admissible solution $\{\lambda_1^{(\kappa)}, \lambda_2^{(\kappa)}\}$, corresponding to a physical eigenstate, satisfying

$$\lim_{\kappa \to 1} \lambda_{1,2}^{(\kappa)} = \pm \eta/2. \tag{A.11}$$

The limiting behaviour of this solution is relatively easily found. Writing

$$\lambda_1^{(\kappa)} = \eta/2 + c_1, \qquad \lambda_2^{(\kappa)} = -\eta/2 + c_2, \qquad \kappa = 1 + \varepsilon, \tag{A.12}$$

we find

$$\sinh^L(c_1 + \eta)\sinh(c_2 - c_1) = (1 + \varepsilon)^L \sinh^L(c_1)\sinh(c_2 - c_1 - 2\eta)$$
$$\sinh^L(c_2)\sinh(2\eta + c_1 - c_2) = (1 + \varepsilon)^L \sinh^L(c_2 - \eta)\sinh(c_1 - c_2). \tag{A.13}$$

These equations imply $\mathcal{O}(c_{1,2}) = \varepsilon$ but also a correction for the difference with $\mathcal{O}(c_1 - c_2) = \varepsilon^L$. The first order correction is found simply from the product of the two equations:

$$c_{1,2} = \varepsilon \frac{\sinh \eta}{\cosh \eta} + \mathcal{O}(\varepsilon^L). \tag{A.14}$$

Using one of the Bethe equations further gives

$$c_1 - c_2 = \varepsilon^L 2 \frac{\sinh(\eta)}{\cosh^{L-1}(\eta)} + \mathcal{O}(\varepsilon^{L+1}). \tag{A.15}$$

Let us investigate $2 \times 2$ Gaudin matrix associated to this state:

$$G = \begin{pmatrix} Lq(\lambda_1) + \varphi(\lambda_{12}) & -\varphi(\lambda_{12}) \\ -\varphi(\lambda_{12}) & Lq(\lambda_1) + \varphi(\lambda_{12}) \end{pmatrix}, \qquad \lambda_{12} = c_1 - c_2. \tag{A.16}$$

Here the functions $q(u)$ and $\varphi(u)$ are those defined in (3.34) and (3.35).

It can be seen that in the $\kappa \to 1$ limit every matrix element will be dominated by $\varphi(\lambda_{12})$ and the leading behaviour is thus

$$G = \varepsilon^{-L} \frac{\cosh^{L-1}(\eta)}{2\sinh(\eta)} \begin{pmatrix} 1 & -1 \\ -1 & 1 \end{pmatrix} + \mathcal{O}(\varepsilon^{-(L-1)}). \tag{A.17}$$

Let us now consider a double integral of the form

$$I^{(\kappa)} = \oint_{\mathcal{C}_1} \frac{du_1}{2\pi} q(u_1) \oint_{\mathcal{C}_2} \frac{du_2}{2\pi} q(u_2) \frac{f(u_1, u_2)}{(\kappa^L P^L(u_1) S(u_1 - u_2) - 1)(\kappa^L P^L(u_2) S(u_2 - u_1) - 1)}, \tag{A.18}$$

where $\mathcal{C}_1$ and $\mathcal{C}_2$ are small contours with radiuses $r_{1,2}$ around the points $\pm \eta/2$, and $f(u_1, u_2)$ is any differentiable function for $u_1 \neq \eta/2, u_2 \neq -\eta/2$. It is important that the radiuses can not be equal, and we investigate the $\kappa \to 1$ limit with fixed $r_1 \neq r_2$.

If $\kappa$ is far enough from 1, then the Bethe roots of the deformed singular solution are outside the contours. However, as we approach $\kappa \to 1$ the roots $\lambda_{1,2}^{(\kappa)}$ necessarily cross the two contours $\mathcal{C}_{1,2}$. The contribution of the double integral can then be evaluated explicitly using the formula (A.8). The end result depends on the ratio of the radiuses $|r_1/r_2|$ and the ratio of the Gaudin matrix elements. It follows from the above that

$$\lim_{\kappa \to 1} \left| \frac{G_{jk}}{G_{lm}} \right| = 1, \quad j, k, l, m = 1, 2. \tag{A.19}$$

Therefore the $\kappa \to 1$ limit of the integral will be zero, for any fixed ratio $|r_1/r_2| \neq 1$.

With this we have shown that the physical singular solutions will not give additional contributions to such integrals as we perform the $\kappa \to 1$ analytic continuation.

# B  The $A$ operator in the $F$-basis

Here we compute the $A$-operator in the F-basis. As it was already remarked in [70], this is easily achieved by using the so-called quantum determinant. We believe that this result is not yet published in the literature, nevertheless the calculation is by no means new, it was already performed by other researchers [73].

The quantum determinant is a specific combination of the $A, B, C,$ and $D$ operators which is proportional to the identity operator. There are in fact four different relations leading to the same scalar factor [2]:

$$A(u+\eta)D(u) - B(u+\eta)C(u) = D(u+\eta)A(u) - C(u+\eta)B(u) =$$

$$= D(u)A(u+\eta) - B(u)C(u+\eta) = A(u)D(u+\eta) - C(u)B(u+\eta) = \prod_{j=1}^{n} b(u-\xi_j) \cdot I. \tag{B.1}$$

All four relations could be used to yield a formula for $A(u)$. The four relations lead to two different expressions for $A$.

**First expression for $A$ in the $F$-basis**

First we choose to compute it using the last relation:

$$A(u) = \left( \prod_{i=1}^{n} b(u, \xi_i) I + C(u)B(u+\eta) \right) D^{-1}(u+\eta). \tag{B.2}$$

Here

$$D^{-1}(u+\eta) = \otimes_{i=1}^{n} \begin{pmatrix} b^{-1}(u+\eta, \xi_i) & \\ & 1 \end{pmatrix}_{[i]}. \tag{B.3}$$

We compute the $CB$ product as

$$C(u)B(u+\eta) = \sum_{i=1}^{n} c(u, \xi_i)c(u+\eta, \xi_i)\sigma_i^+ \sigma_i^- \otimes_{j, j \neq i}$$
$$\begin{pmatrix} b(u, \xi_j)b(u+\eta, \xi_j)b^{-1}(\xi_i, \xi_j) & 0 \\ 0 & b^{-1}(\xi_j, \xi_i) \end{pmatrix}_{[j]} +$$
$$+ \sum_{i, j, i \neq j} c(u, \xi_i)b^{-1}(\xi_i, \xi_j)\sigma_i^+ \otimes c(u+\eta, \xi_j)\sigma_j^- \otimes_{k \neq i, j}$$
$$\begin{pmatrix} b(u, \xi_k)b(u+\eta, \xi_k)b^{-1}(\xi_i, \xi_k) & 0 \\ 0 & b^{-1}(\xi_k, \xi_j) \end{pmatrix}_{[k]}. \tag{B.4}$$

Multiplying this with $D^{-1}$ is relatively straightforward, as $D^{-1}$ is diagonal:

$$C(u)B(u+\eta)D^{-1}(u+\eta) =$$

$$= \sum_{i=1}^{n} c(u,\xi_i)c(u+\eta,\xi_i)\sigma_i^+\sigma_i^- \begin{pmatrix} b^{-1}(u+\eta,\xi_i) & \\ & 1 \end{pmatrix}_{[i]} \otimes_{j,j\neq i}$$

$$\otimes_{j,j\neq i} \begin{pmatrix} b(u,\xi_i)b^{-1}(\xi_i,\xi_j) & \\ & b^{-1}(\xi_j,\xi_i) \end{pmatrix}_{[j]} +$$

$$+ \sum_{i,j,i\neq j} c(u,\xi_i)b^{-1}(\xi_i,\xi_j)\sigma_i^+ \begin{pmatrix} b^{-1}(u+\eta,\xi_i) & \\ & 1 \end{pmatrix}_{[i]} \otimes c(u+\eta,\xi_j)$$

$$\otimes c(u+\eta,\xi_j)\sigma_j^- \begin{pmatrix} b^{-1}(u+\eta,\xi_j) & \\ & 1 \end{pmatrix}_{[j]} \otimes_{k\neq i,j}$$

$$\otimes_{k\neq i,j} \begin{pmatrix} b(u,\xi_k)b^{-1}(\xi_i,\xi_k) & 0 \\ 0 & b^{-1}(\xi_k,\xi_j) \end{pmatrix}_{[k]} =$$

$$= \sum_{i=1}^{n} c^2(u,\xi_i)\sigma_i^+\sigma_i^- \otimes_{j,j\neq i} \begin{pmatrix} b(u,\xi_i)b^{-1}(\xi_i,\xi_j) & \\ & b^{-1}(\xi_j,\xi_i) \end{pmatrix}_{[j]} +$$

$$+ \sum_{i,j,i\neq j} b^{-1}(\xi_i,\xi_j)c(u,\xi_i)\sigma_i^+ \otimes c(u,\xi_j)\sigma_j^- \otimes_{k\neq i,j} \begin{pmatrix} b(u,\xi_k)b^{-1}(\xi_i,\xi_k) & 0 \\ 0 & b^{-1}(\xi_k,\xi_j) \end{pmatrix}_{[k]},$$

where we used twice that

$$c(u+\eta,\xi)b^{-1}(u+\eta,\xi) = c(u,\xi).$$

Adding both terms in (B.2) leads to the first expression for $A$ in the $F$-basis:

$$A_n(u) = \otimes_{i=1}^{n} \begin{pmatrix} b(u,\xi_i)b^{-1}(u+\eta,\xi_i) & 0 \\ 0 & b(u,\xi_i) \end{pmatrix}_{[i]} +$$

$$+ \sum_{i=1}^{n} c^2(u,\xi_i)\sigma_i^+\sigma_i^- \otimes_{j,j\neq i} \begin{pmatrix} b(u,\xi_j)b^{-1}(\xi_i,\xi_j) & 0 \\ 0 & b^{-1}(\xi_j,\xi_i) \end{pmatrix}_{[j]} +$$

$$+ \sum_{i,j,i\neq j} c(u,\xi_i)c(u,\xi_j)b^{-1}(\xi_i,\xi_j)\sigma_i^+ \otimes \sigma_j^- \otimes_{k\neq i,j} \begin{pmatrix} b(u,\xi_k)b^{-1}(\xi_i,\xi_k) & 0 \\ 0 & b^{-1}(\xi_k,\xi_j) \end{pmatrix}_{[k]}.$$

$$\tag{B.5}$$

**Second expression for $A$ in the $F$-basis**

One can also use the relation

$$D(u)A(u+\eta) - B(u)C(u+\eta) = \prod_{j=1}^{n} b(u-\xi_j)\cdot I \tag{B.6}$$

to compute $A$ in closed form. We get

$$A(u) = \prod_{j=1}^{n} b(u-\eta-\xi_j)\cdot D^{-1}(u-\eta) + D^{-1}(u-\eta)B(u-\eta)C(u). \tag{B.7}$$

First we compute $D^{-1}(u-\eta)$:

$$D^{-1}(u-\eta) = \otimes_{i=1}^{n} \begin{pmatrix} b^{-1}(u-\eta-\xi_i) & \\ & 1 \end{pmatrix}_{[i]}. \tag{B.8}$$

Hence the diagonal term:

$$\prod_{j=1}^{n} b(u-\eta-\xi_j) \cdot D^{-1}(u-\eta) = \otimes_{i=1}^{n} \begin{pmatrix} 1 & \\ & b(u-\eta-\xi_i) \end{pmatrix}_{[i]} \tag{B.9}$$

and the non-diagonal part:

$$D^{-1}(u-\eta)B(u-\eta)C(u) = \sum_{i=1}^{n} \begin{pmatrix} b^{-1}(u-\eta-\xi_i) & \\ & 1 \end{pmatrix}_{[i]} c(u-\eta-\xi_i)\sigma_i^- c(u-\xi_i)\sigma_i^+ \otimes_{j,j\neq i}$$

$$\otimes_{j,j\neq i} \begin{pmatrix} b^{-1}(u-\eta-\xi_j)b(u-\eta-\xi_j)b(u-\xi_j)b^{-1}(\xi_i-\xi_j) & \\ & b^{-1}(\xi_j\xi_i) \end{pmatrix}_{[j]} +$$

$$+ \sum_{i,j,i\neq j} \begin{pmatrix} b^{-1}(u-\eta-\xi_i) & \\ & 1 \end{pmatrix}_{[i]} c(u-\eta-\xi_i)\sigma_i^- \begin{pmatrix} b(u-\xi_i)b^{-1}(\xi_i-\xi_j) & \\ & 1 \end{pmatrix}_{[i]} \otimes$$

$$\otimes \begin{pmatrix} b^{-1}(u-\eta-\xi_j) & \\ & 1 \end{pmatrix}_{[j]} \begin{pmatrix} b(u-\eta-\xi_j) & \\ & b^{-1}(\xi_j-\xi_i) \end{pmatrix}_{[j]} c(u-\xi_j)\sigma_j^+ \otimes_{k,k\neq i,j}$$

$$\otimes_{k,k\neq i,j} \begin{pmatrix} b^{-1}(u-\eta-\xi_k)b(u-\eta-\xi_k)b(u-\xi_k)b^{-1}(\xi_j-\xi_k) & \\ & b^{-1}(\xi_k-\xi_i) \end{pmatrix}_{[k]}. \tag{B.10}$$

Using the identity $c(u-\eta-\xi_i) = c(u-\xi_i)b^{-1}(u-\xi_i)$ we obtain (4.27).

## C  Further details on the two particle propagator

### C.1  The Tr $\check{B}\check{C}\check{C}\check{B}$ case

Here we treat the two-particle propagator in the case of $c < a < b < d$, when the trace to be computed is Tr $\check{B}\check{C}\check{C}\check{B}$. The following contracted traces are non-vanishing:

$$\text{Tr } B^{(\ell_1)}C^{(\ell_2)}C^{(\ell_3)}B^{(\ell_4)}{}_{(iijj)} = \sum_{i,j,i\neq j} c_i^2 c_j^2 \, b_{ij}^{-1} b_{ij}^{-1} \, b_i^{\ell_2} b_j^{\ell_1+\ell_2+\ell_3+2} \prod_{k,k\neq i,k\neq j} b_k^L b_{ik}^{-1} b_{jk}^{-1} + b_{ki}^{-1} b_{kj}^{-1}$$

$$\text{Tr } B^{(\ell_1)}C^{(\ell_2)}C^{(\ell_3)}B^{(\ell_4)}{}_{(ijij)} = \sum_{i,j,i\neq j} c_i^2 c_j^2 \, b_{ij}^{-1} b_{ji}^{-1} \, b_i^{\ell_2+\ell_3+1} b_j^{\ell_1+\ell_2+1} \prod_{k,k\neq i,k\neq j} b_k^L b_{ik}^{-1} b_{jk}^{-1} + b_{ki}^{-1} b_{kj}^{-1}. \tag{C.1}$$

These leads to the following $f$ function (the $g_k$ and $h_k$ functions are the same):

$$f(u_1,u_2) = f_{non-sym.}^{(\ell_2,\ell_1+\ell_2+\ell_3+2)}(u_1,u_2) + f_{sym.}^{(\ell_2+\ell_3+1,\ell_1+\ell_2+1)}(u_1,u_2). \tag{C.2}$$

Using this $f$ function in the contour integral, in the summation over residues the $\sum_i \ldots$ term is canceled, as $f_{sym.}$ and $f_{non-sym.}$ cancels each other. This case is very similar to the detailed Tr $\check{A}\check{C}\check{B}$ case with the main difference, that in this case, $f$ is free of poles. Hence the final expression:

$$\langle a,b|e^{-iHt}|c,d\rangle = \oint_{\mathcal{C}_\pm \times \mathcal{C}_\pm} \frac{du_1 du_2}{(2\pi i)^2} \left( S(u_2-u_1)P^{a-c}(u_1)P^{L+b-d}(u_2) + P^{b-c}(u_1)P^{L+a-d}(u_2) \right) \times$$

$$\times q(u_1)q(u_2) \frac{\exp(-i(\varepsilon(u_1)+\varepsilon(u_2))t)}{(1-P^L(u_1)S(u_1-u_2))(1-P^L(u_2)S(u_2-u_1))}. \tag{C.3}$$

## C.2 The Tr $\check{A}\check{B}\check{C}$ case

Here we treat the two-particle propagator in the case of $a = c < d < b$, when the trace to be computed is Tr $\check{A}\check{B}\check{C}$. The following contracted traces are non-vanishing:

$$
\text{Tr } A^{(\ell_1)}B^{(\ell_2)}C^{(\ell_3)}{}_{(()ii)} = \sum_{i=1}^{n} c_i^2 b_{i0}^{-1} b_i^{\ell_3} \prod_{j,j\neq i}\left(b_j^L b_{0j}^{-1} + b_{j0}^{-1}b_{ji}^{-1}\right)
$$

$$
\text{Tr } A^{(\ell_1)}B^{(\ell_2)}C^{(\ell_3)}{}_{(iiii)} = \sum_{i=1}^{n} c_i^4 b_i^{\ell_3-1} \prod_{j,j\neq i}\left(b_j^L b_{ij}^{-2} + b_{ji}^{-2}\right)
$$

$$
\text{Tr } A^{(\ell_1)}B^{(\ell_2)}C^{(\ell_3)}{}_{(iijj)} = \sum_{i,j,i\neq j} c_i^2 c_j^2 b_{ij}^{-1} b_{ji}^{-1} b_i^{-1} b_j^{\ell_3} \prod_{k,k\neq i,j}\left(b_k^L b_{ik}^{-1}b_{jk}^{-1} + b_{ki}^{-1}b_{kj}^{-1}\right)
$$

$$
\text{Tr } A^{(\ell_1)}B^{(\ell_2)}C^{(\ell_3)}{}_{(ijji)} = \sum_{i,j,i\neq j} c_i^2 c_j^2 b_{ij}^{-1} b_{ji}^{-1} b_i^{\ell_2+\ell_3+1} b_j^{\ell_3+\ell_1+1} \prod_{k,k\neq i,j}\left(b_k^L b_{ik}^{-1}b_{jk}^{-1} + b_{ki}^{-1}b_{kj}^{-1}\right).
$$

(C.4)

This case is similar to the detailed Tr $\check{B}\check{C}\check{B}\check{C}$ case, with one further term (Tr $\check{A}\check{B}\check{C}_{(()ii)}$). The corresponding $f$ function is the following:

$$
f(u_1,u_2) = f_{sym.}^{(-1,\ell_3)}(u_1,u_2) + f_{sym.}^{(\ell_2+\ell_3+1,\ell_3+\ell_1+1)}(u_1,u_2).
$$

(C.5)

Due to the presence of $f_{sym.}^{(-1,\ell_3)}$, $f$ is singular, and the residue of $f$ corresponds to the extra term Tr $\check{A}\check{B}\check{C}_{(()ii)}$. This leads to the final expression:

$$
\begin{aligned}
\langle a,b|e^{-iHt}|c,d\rangle = &\oint_{\mathcal{C}_\pm\times\mathcal{C}_\pm}\oint \frac{du_1 du_2}{(2\pi i)^2}\left(P^{b-d}(u_2) + P^{b-c}(u_1)P^{L+a-d}(u_2)\right) \times \\
&\times q(u_1)q(u_2)\frac{\exp(-i(\varepsilon(u_1)+\varepsilon(u_2))t)}{(1-P^L(u_1)S(u_1-u_2))(1-P^L(u_2)S(u_2-u_1))}.
\end{aligned}
$$

(C.6)

## C.3 The Tr $\check{A}\check{A}$ case

Here we treat the two-particle propagator in the case of $a = c < b = d$, when the trace to be computed is Tr $\check{A}\check{A}$. This case has the most non-vanishing contracted traces, namely the following ones:

$$
\text{Tr } A^{(\ell_1)}A^{(\ell_2)}{}_{(()())} = \prod_{i=1}^{n}\left(b_i^{\ell_1+\ell_2} + b_{i0}^{-2}\right) = \prod_{i=1}^{n}\left(b_i^L b_{0i}^{-2} + b_{i0}^{-2}\right)
$$

$$
\text{Tr } A^{(\ell_1)}A^{(\ell_2)}{}_{(()ii)} = \sum_{i=1}^{n} c_i^2 b_{i0}^{-1} b_i^{-1} \prod_{j,j\neq i}\left(b_j^L b_{0j}^{-1}b_{ij}^{-1} + b_{j0}^{-1}b_{ji}^{-1}\right)
$$

$$
\text{Tr } A^{(\ell_1)}A^{(\ell_2)}{}_{(ii())} = \sum_{i=1}^{n} c_i^2 b_{i0}^{-1} b_i^{-1} \prod_{j,j\neq i}\left(b_j^L b_{0j}^{-1}b_{ij}^{-1} + b_{j0}^{-1}b_{ji}^{-1}\right)
$$

$$
\text{Tr } A^{(\ell_1)}A^{(\ell_2)}{}_{(iiii)} = \sum_{i=1}^{n} c_i^4 b_i^{-2} \prod_{j,j\neq i}\left(b_j^L b_{ij}^{-2} + b_{ji}^{-2}\right)
$$

$$
\text{Tr } A^{(\ell_1)}A^{(\ell_2)}{}_{(iijj)} = \sum_{i,j,i\neq j} c_i^2 c_j^2 b_{ij}^{-1} b_{ji}^{-1} b_i^{-1} b_j^{-1} \prod_{k,k\neq i,j}\left(b_k^L b_{ik}^{-1}b_{jk}^{-1} + b_{ki}^{-1}b_{kj}^{-1}\right)
$$

$$
\text{Tr } A^{(\ell_1)}A^{(\ell_2)}{}_{(ijji)} = \sum_{i,j,i\neq j} c_i^2 c_j^2 b_{ij}^{-1} b_{ji}^{-1} b_i^{\ell_2} b_j^{\ell_1} \prod_{k,k\neq i,j}\left(b_k^L b_{ik}^{-1}b_{jk}^{-1} + b_{ki}^{-1}b_{kj}^{-1}\right).
$$

(C.7)

This case is similar to the previous ones, however, it brings a minor novelty: the $\text{Tr}\,\breve{A}\breve{A}_{(()())}$ contracted trace is new. The $f$ function is the following:

$$f(u_1, u_2) = f_{sym.}^{(-1,-1)}(u_1, u_2) + f_{sym.}^{(\ell_2, \ell_1)}(u_1, u_2). \tag{C.8}$$

As we expect, due to $f_{sym.}^{(-1,-1)}$, $f$ has poles at $u_i = 0$, $i = 1, 2$. This leads to the following further modification of the residue equation:

$$\oint_{\mathcal{C} \times \mathcal{C}} \oint \frac{du_1 du_2}{(2\pi\mathrm{i})^2} f(u_1, u_2) \prod_i \frac{h_i(u_1, u_2)}{g_i(u_1) g_i(u_2)} =$$

$$= \sum_i f(\xi_i, \xi_i) h_i(\xi_i, \xi_i) \left( \text{Res}_{u_1=\xi_i} \frac{1}{g_i(u_1)} \right) \left( \text{Res}_{u_2=\xi_i} \frac{1}{g_i(u_2)} \right) \prod_{j, j \neq i} \frac{h_j(\xi_i, \xi_i)}{g_j(\xi_i) g_j(\xi_i)} +$$

$$+ \sum_{i,j,i \neq j} f(\xi_i, \xi_j) \frac{h_j(\xi_i, \xi_j)}{g_j(\xi_i)} \left( \text{Res}_{u_2=\xi_j} \frac{1}{g_j(u_2)} \right) \frac{h_i(\xi_i, \xi_j)}{g_i(\xi_j)} \left( \text{Res}_{u_1=\xi_i} \frac{1}{g_i(u_1)} \right) \prod_{k, k \neq i, k \neq j} \frac{h_k(\xi_i, \xi_j)}{g_k(\xi_i) g_k(\xi_j)} +$$

$$+ \sum_{i=1}^n \left( \text{Res}_{u_1=\tilde{z}} f(u_1, \xi_i) \right) \frac{h_i(\tilde{z}, \xi_i)}{g_i(\tilde{z})} \left( \text{Res}_{u_2=\xi_i} \frac{1}{g_i(u_2)} \right) \prod_{j, j \neq i} \frac{h_j(\tilde{z}, \xi_i)}{g_j(\tilde{z}) g_j(\xi_i)} +$$

$$+ \sum_{i=1}^n \left( \text{Res}_{u_2=\tilde{z}} f(\xi_i, u_2) \right) \frac{h_i(\xi_i, \tilde{z})}{g_i(\tilde{z})} \left( \text{Res}_{u_1=\xi_i} \frac{1}{g_i(u_1)} \right) \prod_{j, j \neq i} \frac{h_j(\xi_i, \tilde{z})}{g_j(\xi_j) g_j(\tilde{z})} +$$

$$+ \left( \text{Res}_{u_1=\tilde{z}, u_2=\tilde{z}} f(u_1, u_2) \right) \prod_{i=1}^n \frac{h_i(\tilde{z}, \tilde{z})}{g_i(\tilde{z}) g_i(\tilde{z})}.$$

Note, that due to the product structure in all the relevant places, every residue can be taken successively.

The different parts of the partition function are given as

$$\begin{aligned}
\text{Tr}\,\breve{A}\breve{A}_{(iiii)} &= \sum_i f(\xi_i, \xi_i) h_i(\xi_i, \xi_i) \prod_{j, j \neq i} \frac{h_j(\xi_i, \xi_i)}{g_j(\xi_i) g_j(\xi_i)} \\
\text{Tr}\,\breve{A}\breve{A}_{(()ii)} &= \sum_{i=1}^n \left( \text{Res}_{u_1=0} f(u_1, \xi_i) \right) \frac{h_i(0, \xi_i)}{g_i(0)} \prod_{j, j \neq i} \frac{h_j(0, \xi_i)}{g_j(0) g_j(\xi_i)} \\
\text{Tr}\,\breve{A}\breve{A}_{(ii())} &= \sum_{i=1}^n \left( \text{Res}_{u_2=0} f(\xi_i, u_2) \right) \frac{h_i(\xi_i, 0)}{g_i(0)} \prod_{j, j \neq i} \frac{h_j(\xi_i, 0)}{g_j(\xi_j) g_j(0)} \\
\text{Tr}\,\breve{A}\breve{A}_{(()())} &= \left( \text{Res}_{u_1=0, u_2=0} f(u_1, u_2) \right) \prod_{i=1}^n \frac{h_i(0,0)}{g_i(0) g_i(0)}.
\end{aligned} \tag{C.9}$$

Hence, the final expression follows similarly as in all the previous cases:

$$\langle a, b | e^{-\mathrm{i}Ht} | c, d \rangle = \oint_{\mathcal{C}_\pm \times \mathcal{C}_\pm} \oint \frac{du_1 du_2}{(2\pi\mathrm{i})^2} \left( 1 + P^{b-c}(u_1) P^{L+a-d}(u_2) \right) \times$$

$$\times q(u_1) q(u_2) \frac{\exp(-\mathrm{i}(\varepsilon(u_1) + \varepsilon(u_2))t)}{(1 - P^L(u_1) S(u_1 - u_2))(1 - P^L(u_2) S(u_2 - u_1))}. \tag{C.10}$$

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
