# Peer review of "The propagator of the finite XXZ spin-$\tfrac{1}{2}$ chain"

_SciPost Physics, doi:SciPost Phys. 6, 063 (2019)_

## Round 2 · Referee Report · Anonymous (Referee 1) · 2018-11-7

Strengths

1- The paper presents an original way to compute the propagator by means of an ingenious use of the F-basis in the mirror channel.

2- The paper is clearly written and computations are given in detail (at least in the cases of 1 or 2 particles which are explicitly treated here).

Weaknesses

1- The proof of the main result (formula for the multi-particle propagator) in not given in the paper. The authors write that they have "constructed a general proof of this result, which will be presented elsewhere due to its length and technical nature". In fact, the computations become already so cumbersome in the two-particle case that you may wonder whether it is really tractable in the general case, and you have to believe the authors when they say they indeed did it.

2- The authors seem to be unaware of the existing literature concerning dynamical correlation functions that would be especially relevant for the subject of their paper. In particular, they seem to be unaware of the following reference:

N. Kitanine, J. M. Maillet, N. A. Slavnov, V. Terras, "Dynamical correlation functions of the XXZ spin-1/2 chain", Nucl.Phys. B729 (2005) 558-580, arXiv:hep-th/0407108.

in which a not so different approach was developed for the computation of a slightly different object (a time-dependent correlation function), leading to a very similar result in its form. In particular, the result of the present paper can be easily predicted in the context of the aforementioned reference.

Report

The aim of the paper is the computation of the propagator of the finite XXZ spin-1/2 chain, namely of the matrix elements of the time evolution operator in the local spin basis of the model.
To perform this computation, the authors use a Trotter approximation of the time evolution operator as a product of transfer matrices. The resulting matrix elements are then written as a trace of products of elements of the so-called quantum transfer matrix, and evaluated in the F-basis, a basis in which the explicit expression of these quantum transfer matrix elements take very simple form.

I find this paper interesting due to the ingenious use of the F-basis in the framework of the quantum transfer matrix. I have however two main concerns.

The first one is that I was quite disappointed that the authors did not present the general proof of their main result in the paper. In fact, they only proved explicitly the formula in the cases with one or two particles only. The computations in the two particle case are already quite involved, so that it seems not so obvious that it is still manageable in the general case. The authors pretend they have done it, and I really would have liked to see the proof here.

The second one is that there already exists in the literature a paper, not even cited here:

N. Kitanine, J. M. Maillet, N. A. Slavnov, V. Terras, "Dynamical correlation functions of the XXZ spin-1/2 chain", Nucl.Phys. B729 (2005) 558-580, arXiv:hep-th/0407108,

in which the propagator of the finite spin chain was summed up in the form of a multiple integral of the same form (i.e. on the same contour, also with the Bethe equations in the denominator), although in a slightly different context (for the study of time-dependent correlation functions). The authors of the aforementioned reference showed moreover that it is possible to obtain this result in two different ways: from a direct computation using a Trotter decomposition (see formulas (1.6), (4.1) and (4.2) there, which are exactly equivalent to formula (2.35) of the present paper), or from the insertion of a complete set of eigenstates (with deformation by a twist so as to effectively ensure the completeness), an option which is discussed by the authors of the present paper, but ruled out as too complicated. In fact, if one uses the latter option (following the procedure described in section 5 of the above 2005 paper), one obtains straightforwardly a multiple integral representation of the propagator that should be equivalent to the result presented here. The numerous discussions in the present paper of whether one can or not obtain the result by a sum over the Bethe roots clearly show that the authors are not aware of this previous work on the subject.

Requested changes

Major changes:

1- Since the authors claim they already constructed the general proof of their result by their method, I would have liked to see it in the present paper. I understand that it may be a bit long but, as anyway they want to publish this proof, I think this should be done here, at least in a sketchy way. In fact, since the result in itself is not a surprise (considering that one can obtain the multiple integral representation directly by insertion of a complete set of states, following the scheme presented in the 2005 reference I mentioned above), the main interest of the present paper is the inventive method presented here. It should therefore be shown that it is manageable in the general case as well, and not only for 1 or 2 particles. The authors may use appendices for this if they do not want to make the paper too heavy.

2- The authors should include the aforementioned reference to the 2005 paper of N. Kitanine et al., and modify accordingly their discussion, for instance before section 2.1, around formula (2.35), in section 4.4, in section 5.3, in section 6 and of course in introduction and conclusion. The result obtained by the two methods should of course coincide.

Other minor changes:

3- It would be nice to have explicit references in the first paragraph of introduction, concerning "the study of the state functions and correlation functions in the ground state or at finite temperature", concerning "the study of out-of-equilibrium situations" (even if in that case the references are presented later), and concerning "experimental advances that make it possible to measure the dynamical properties of isolated quantum systems".

4- Still in the introduction, on page 2, I think that the sentence "It was already argued that even time dependent local correlators could be computed within the QTM, somewhat analogous to the determination of finite temperature correlators" would deserve, among other references to the computation of correlation functions within QTM, a reference to the work of Sakai:

K. Sakai, "Dynamical correlation functions of the XXZ model at finite temperature", J. Phys. A 40:7523-7542 (2007), arXiv:cond-mat/0703319.

5- Just before formula (2.19), the authors should also cite

M. Gaudin, B. M. McCoy, and T. T. Wu, "Normalization sum for the Bethe's hypothesis wave functions of the Heisenberg-Ising chain" Phys. Rev. D 23 (1981).

6- There is an unfinished sentence at the end of section 4 that should be completed.

---

## Round 3 · Referee Report · Anonymous (Referee 1) · 2019-5-2

Strengths

1- The author have followed my previous recommendation to refer to what is now Reference [21], and have built their main proof of the integral representation of the propagator by closely following the procedure given in this reference. This indeed provides a much more straightforward derivation of this representation.

Weaknesses

1- The authors have not given the general proof of the main result by means of their original method using the F-basis. As I was fearing from the two-particle case, it seems that this is not really tractable in the general case. This is a pity since the method was a priori original and ingenious, but I don't insist on this.

2- References on the previous literature are still sometimes treated in a rushed manner.

3- One may wonder whether the obtained formula is really useful: is it really possible, from the final result, i.e. from the multiple integral representation obtained for the propagator, to study the large-size and long-time limits ? It seems to me quite complicated. Some attempts have already been made to study the asymptotic behavior of these kinds of integral representations in the context of correlation functions at equilibrium (based on the work of references [20,21]). In particular, in the couple of references:

  • N. Kitanine, K.K. Kozlowski, J.M. Maillet, N.A. Slavnov, V. Terras, "Algebraic Bethe ansatz approach to the asymptotic behavior of correlation functions", Journal of Statistical Mechanics: Theory and Experiment, P04003 (2009)

  • N. Kitanine, K.K. Kozlowski, J.M. Maillet, N.A. Slavnov, V. Terras, "Riemann–Hilbert Approach to a Generalised Sine Kernel and Applications", Commun. Math. Phys. (2009) 291: 691

the large-distance asymptotic expansion was derived in the static case from the multiple integral representation of Reference [20]. But the derivation is quite complicated, and I don't know any work that managed to extract a long-time and large-distance asymptotic behavior directly from the representation obtained in reference [21].

Report

In this revised version, the authors have followed my suggestion to consider the proof of their result through reference [21], so that they now present a short derivation of their main formula. They have moreover added an illustration of their result by presenting a multiple integral representation for the Loschmidt amplitude for the domain wall quench. Unfortunately, the proof by means of the F-basis is still limited to the two-particle case, but it seems to be too complicated to be presented in the general case. The question remains to see whether the integral representation for the propagator is really useful for effective calculations. I think it would at least deserve some more critical discussion, based on what it was possible to do from similar representations in the context of the study of correlation functions.

Requested changes

I think that, with respect to the previous version, the paper has improved, and is now suitable to be published, provided the authors consider the following small changes:

1- Add a few references at some places:

1.1- For instance, in the very beginning, I think the sentence "whereas the largest part of the literature is devoted to the study of the state functions and correlation functions in the ground state or at finite temperatures" would deserve several more references:

  • some references about the q-vertex operator approach of the Kyoto group, at least of the book "Algebraic analysis in solvable lattice models" of Jimbo and Miwa.

  • some references about the ABA approach of the Lyon group, for instance:

N Kitanine, JM Maillet, V Terras, "Correlation functions of the XXZ Heisenberg spin-1/2 chain in a magnetic field", Nuclear Physics B 567, 554-582 (2000)

and some further papers, such as references [20,21]

  • some references about the approach of Boos, Jimbo, Miwa, Smirnov and Takeyama, for instance:

H. Boos, M. Jimbo, T. Miwa, F. Smirnov, Y. Takeyama, "Algebraic representation of correlation functions in integrable spin chains", Annales Henri Poincare 7: 1395-1428 (2006)

M. Jimbo, T. Miwa, F. Smirnov, "Hidden Grassmann Structure in the XXZ Model III: Introducing Matsubara direction", J. Phys. A 42: 304018 (2009)

and some other papers

  • some references about the QTM approach of Goehmann, Klumper et al, for instance:

F. Göhmann, A. Klümper, A. Seel, "Integral representations for correlation functions of the XXZ chain at finite temperature", J.Phys. A37 (2004) 7625-7652

and some further papers

  • some references about the analytic derivation of the asymptotic behavior of correlation functions, for instance

N. Kitanine, K.K. Kozlowski, J.M. Maillet, N.A. Slavnov, V. Terras, "Algebraic Bethe ansatz approach to the asymptotic behavior of correlation functions", Journal of Statistical Mechanics: Theory and Experiment, P04003 (2009)

N. Kitanine, K.K. Kozlowski, J.M. Maillet, N.A. Slavnov, V. Terras, "A form factor approach to the asymptotic behavior of correlation functions in critical models", Journal of Statistical Mechanics: Theory and Experiment 2011 (12), P12010

as well as the review they already provided.

1.2- Also, when introducing the Dzyaloshinsky-Moriya interaction term, the authors do not provide any reference at all about previous literature on the subject. In addition to the seminal papers, they could also cite:

F. C. Alcaraz and W. F. Wreszinski, "The Heisenberg XXZ Hamiltonian with Dzyaloshinsky-Moriya Interactions", Journal of Statistical Physics, Vol. 58, 45 (1990)

1.3- What the authors call the "Izergin-Korepin determinant" was first derived by Izergin alone in

A.G. Izergin, Sov. Phy. Dokl. 32 878-9 (1987).

This reference should be cited.

1.4- The appendix on multidimensional residues also deserves some reference.

2- In the conclusion, I would like to see a more critical discussion about the possibility to extract in practice the large-size limit, finite-size effects, and long-time limit from such multiple integral representations. As mentioned in the paragraph "Weaknesses", point 3, previous attempts in this direction from similar formulas in the context of correlation functions have shown that it was not so easy. It was shown to be possible in the static case (from the formulas of reference [20]), and I think the works

  • N. Kitanine, K.K. Kozlowski, J.M. Maillet, N.A. Slavnov, V. Terras, "Algebraic Bethe ansatz approach to the asymptotic behavior of correlation functions", Journal of Statistical Mechanics: Theory and Experiment, P04003 (2009)

  • N. Kitanine, K.K. Kozlowski, J.M. Maillet, N.A. Slavnov, V. Terras, "Riemann–Hilbert Approach to a Generalised Sine Kernel and Applications", Commun. Math. Phys. (2009) 291: 691

should be mentioned in that context. However, in the dynamical case, it seems to me that the question still remains open.

3- Finally, since the core of the paper is now based on the proof issued from reference [21] which I have explicitly pointed to the attention of the authors in my previous report, I think it would be correct that the authors thank me for that in the acknowledgments (as the "anonymous referee").

---

## Round 3 · Author Response

We are thankful to the referee for the review of the manuscript and the comments. The two main points of critique of the referee were that 1. We did not mention/use an earlier result of Kitanine et. al. (hep-th/0407108) 2. We did not publish the full proof of our formulas, just up to two particles.

We certainly agree with the first point. We did not know the particular paper pointed out by the referee, and indeed the lemmas and theorems there are very useful for our purposes too. Therefore, in addition to properly citing this paper we also included a new derivation of our results using the methods of this paper. So we present now our paper after a major revision.

With regard to the second point we do not completely agree. The full combinatorial proof is long and technical and we did not find an easy way to present it. In our own notes it takes more than 40 pages. And now, with the much more simple proof using the twisted transfer matrix and the kappa->0 limit we do not see reasons to present the considerably longer proof as well.

Furthermore, one could argue that the second part, using the F-basis could be deleted completely from the manuscript. However, we chose to keep it, in order to show an alternative idea, which might be still useful in other circumstances, for other derivations or other models.

---

## Round 3 · List of Changes

-We included references to the earlier works dealing with dynamical correlation functions.
-We restructured the manuscript, and included a complete derivation using the eigenstate basis of the twisted transfer matrix.
-We explained that the kappa deformation is not only a neat mathematical trick, but it can also be used to compute the propagator for the XXZ model perturbed by a Dzyaloshinskii–Moriya interaction term.
-We added a new section: as an application we computed the finite volume Loschmidt echo for the quench from the domain wall state. This exact result can be the starting point of later asymptotic analysis.
-We added some references requested by the referee.

---

## Round 4 · Referee Report · Anonymous (Referee 1) · 2019-5-20

Report

All suggested corrections have been made, and I think that the paper is now in a suitable form to be published.

---

## Round 4 · Author Response

We are thankful to the referee for the comments, we implemented the requests.

---

## Round 4 · List of Changes

1. We added all the requested references. (in the case of the DM interaction term we only added the paper dealing with the XXZ case)

  2. We added a criticial discussion in the Conclusions, this is the last paragraph.

  3. We added an acknowledgement of the useful help of the referee. Indeed this is appropriate in the present case, we should have added this already in the last version.

---

## Editorial Decision

published